# Relevance of feedbacks between water availability and crop systems using a coupled hydrological – crop growth model

Sneha Chevuru[1], Rens L.P.H. van Beek[1], Michelle T.H. van Vliet[1], Jerom P.M. Aerts [2&3], Marc F.P. Bierkens[1&4]

Department of Physical Geography, Utrecht University, The Netherlands.
Water Resources Section, Faculty of Civil Engineering and Geosciences, Delft University of Technology, The Netherlands
Department of Hydraulic Engineering, Faculty of Civil Engineering and Geosciences, Delft University of Technology, The Netherlands
Unit Subsurface & Groundwater Systems, Deltares, Utrecht, The Netherlands

*Correspondence to*: Sneha Chevuru (s.chevuru@uu.nl )

## Abstract

Individual hydrological and crop growth models often oversimplify underlying processes, reducing the accuracy of both simulated hydrology and crop growth dynamics. While crop models tend to generalize soil moisture processes, most hydrological models commonly use constant vegetation parameters and prescribed phenologies, neglecting the dynamic nature of crop growth. Despite some studies that have coupled hydrological and crop models, a limited understanding exists regarding the feedbacks between hydrology and crop growth. Our objective is to quantify the feedback between crop systems and hydrology on a fine-grained spatio-temporal level. To this end, the PCR-GLOBWB 2 hydrological model was coupled with the WOFOST crop growth model to quantify both the one-way and two-way interactions between hydrology and crop growth on a daily timestep and at 5 arc minutes (~10 km) resolution. Our study spans the Contiguous United States (CONUS) region and covers the period from 1979 to 2019, allowing a comprehensive evaluation of the feedback between hydrology and crop growth dynamics. We compare individual (stand-alone) as well as one-way and two-way coupled WOFOST and PCR-GLOBWB 2 model runs and evaluate the average crop yield and its interannual variability for rainfed and irrigated crops as well as simulated irrigation water withdrawal for maize, wheat and soybean. Our results reveal distinct patterns in the temporal and spatial variation of crop yield depending on the included interactions between hydrology and crop systems. Evaluating the model results against reported yield and water use data demonstrates the efficacy of the coupled framework in replicating observed irrigated and rainfed crop yields. Our results show that two-way coupling, with its dynamic feedback mechanisms, outperforms one-way coupling for rainfed crops. This improved performance stems from the feedback of WOFOST crop phenology to the crop parameters in the hydrological model. Our results suggest that when crop models are combined

with hydrological models, a two-way coupling is needed to capture the impact of interannual
climate variability on food production.

## 1 Introduction

Global trends in population and economic growth are expected to increase the demand for
water, food, and energy, threatening the sustainable and equitable use of natural resources
(Sophocleous, 2004; Tompkins and Adger, 2004). Water as a resource plays a crucial role in
crop growth, cooling of thermoelectric plants, hydropower generation, and covering domestic
and industrial demand. Water, therefore, is an essential resource at the core of the Water-
Energy-Food-Ecosystem (WEFE) nexus. Currently, 70% of total global freshwater
withdrawals are accounted for by agriculture, making it the largest water user among all sectors
(Dubois, 2011). The Food and Agriculture Organization (FAO) of the United Nations estimated
that the demand for water and food resources will likely increase by 50% by 2050 compared
to 2015 (IRENA, 2015; Corona-López et al., 2021). The increasing demand for water and food
will likely have negative impacts on the environment and will inhibit socio-economic
development if a gap opens between growing water demand and water availability.
The critical interplay between hydrology and crop growth becomes evident during
hydroclimatic extremes (e.g. droughts, heatwaves), as rising demands coincide with potential
declines in both water resources and food production (crop yield) (Jackson et al., 2021). In
addressing the complexities associated with these challenges, studies by Jägermeyr et al.
(2017), utilizing a dynamic vegetation model (LPJmL), evaluated achievable irrigated crop
production under sustainable water management. Their findings revealed that 41% of global
water use currently compromises environmental flow requirements crucial for river
ecosystems, potentially leading to losses in irrigated croplands. Concurrently, research by
Vörösmarty et al. (2000) and Leclère et al., (2014) projects the impacts of climate change on
global agricultural systems, foreseeing an increase in irrigated areas in the future, underscoring
the necessity for significant investments in irrigation, energy, and water resource management.
These findings emphasize the urgent need for improved modelling approaches to assess the
complex interaction between water availability, climate change, and crop yields.
To address these challenges, biophysical process-based models have been widely used to study
the interactions between hydrology and crop growth (Siad et al., 2019; Zhang et al., 2021).
These models provide valuable insights into how meteorological events influence water
availability for crops, as well as how changes in crop growth and senescence affect
hydrological fluxes such as evapotranspiration and root water uptake. However, existing stand-
alone crop models and hydrological models often simplify these processes.  For instance, crop
models usually incorporate a simplified soil-water balance (Zhang et al., 2021) that overlooks
local hydrological processes and often do not account for water use for irrigation and non-
agricultural sectors. Conversely, most hydrological models simplify or neglect the effects of
land cover, phenology, and vegetation changes on hydrological fluxes and the state of available
water resources (Tsarouchi et al., 2014). These simplifications arise due to computational
expediency, disparities in process scales between hydrology at the river basin level and crop
yield at the field level, incomplete understanding of the other domain by model developers, or
because of epistemological uncertainty (Siad et al. 2019; McMillan et al., 2018; Shafiei et al.
2014). Recognizing the strengths of both crop models and hydrological models, a coupling
allows for the exploration of dynamic crop growth's influence on hydrology and water use.
Additionally, a model coupling allows the incorporation of spatio-temporal variations in
hydrological fluxes, including water use, in estimates of crop yield. This understanding
becomes crucial when assessed at the regional to global scale, where local deficits can have
cascading consequences for both water and food security at the basin scale.
The rationale for coupling hydrological and crop growth models is twofold. First, coupling
these models allows for the possibility to assess the impact of limited irrigation water
availability on crop yield. Second, it enables a detailed analysis of how changes in crop type
and growth stages influence groundwater and surface water availability, particularly through
processes such as evapotranspiration and root water uptake. By combining a hydrological
model with a crop growth model, this study aims to enhance our understanding of hydrological
and crop growth interactions and their implications for agricultural productivity and water
resource management on the continental scale.
Previous studies have attempted to couple hydrological and crop models. Noteworthy efforts
by Droppers et al. (2021) have successfully coupled hydrological and crop models, primarily
focusing on achieving attainable crop production. However, these efforts were conducted at
half-degree (~50 km) spatial resolution and focused on long-term average crop yield. They
therefore fall short in exploring the aspects of fine-scale spatiotemporal variability in particular
as a result of interannual climate variability. Other recent efforts to couple crop growth models
and global hydrological models (Jägermeyr et al., 2017) predominantly focus on assessing
yield under different scenarios or adaptation measures. However, limited work focused on
delving into how two-way interactions and feedback mechanisms between crop growth and
hydrological systems operate.
In addition, integrated assessment models have been instrumental in studying the combined
effects of climate change and socio-economic developments on crop yield and water resources
at a large scale. Typically, these models operate on a macro-regional level (Easterling, 1997)
and use annual (or 5 to 10 yearly timesteps), neglecting the impacts of inter- and intra-annual
variability and particularly short-term hydroclimatic extremes. Furthermore, integrated
assessment models often adopt an optimization modelling approach, making them less suitable
for studying the effects of hydroclimatic extremes (Ewert et al., 2015).
Another class of efforts to link water to crop production are water-food nexus studies, that,
however, tend to concentrate on local linkages or provide qualitative descriptions of existing
connections (Momblanch et al., 2019). For instance, a recent review of water-food nexus
studies focusing on the contiguous United States (CONUS), shows that such studies focus
mainly on water security indicators (Veettil et al., 2022) or climate variability impacts on crop
yields (Huang et al., 2021). However, knowledge gaps persist, as water and food resources are
often evaluated separately (Corona-López et al., 2021), exploring allocations through an
optimization model (Mortada et al., 2018) that lacks spatiotemporal variability considerations.
Notably, there is a lack of effort to understand the interactions between hydrology and crop
growth. Further research is needed to bridge these gaps and enhance our understanding of the
dynamic and interlinked processes shaping the water-food nexus.
To address this knowledge gap, this study aims to quantify the two-way interactions between
crop growth and hydrology, hypothesizing that coupling a crop growth model with a
hydrological model will improve both crop yield and hydrological predictions by incorporating
dynamic feedbacks between water availability and crop processes. Specifically, we
hypothesize that: (1) a more realistic representation of soil moisture dynamics and water
availability will lead to better estimates of water stress and yield; and that (2) directly
integrating crop growth information into hydrological models will enhance the accuracy of
predictions regarding irrigation needs and water resource allocation. To test these hypotheses,
we compare three modeling approaches: a stand-alone crop model, a one-way coupled model
(where hydrological conditions influence crop growth but not vice versa), and a two-way
coupled model (where interactions between hydrology and crop growth are fully represented).
By evaluating these different approaches, we aim to determine whether dynamic hydrological-

crop growth feedbacks improve the performance of crop yield and irrigation water use simulations.

Although this study has a  global scale in scope, we limit this analysis to the Contiguous United States (CONUS) region, to keep the analysis tractable and because CONUS has detailed information on yearly crop production and water use. CONUS is a major producer and contributor to the global production of three primary crops: maize, soybean, and wheat. These crops were selected due to their substantial impact on the agricultural landscape and their pivotal role in shaping global food production trends. The CONUS serves as an ideal study area owing to its extensive availability of relevant data, particularly on agricultural statistics and irrigation water withdrawals, which can provide a basis for analysis and model evaluation. Additionally, the CONUS region exhibits diverse climatic and geographic conditions, contributing to a better understanding of crop and water system dynamics and their responses to various environmental factors.

To test the hypotheses coined above, the PCR-GLOBWB 2 hydrological model (Sutanudjaja et al., 2018) is coupled to the WOFOST crop model (de Wit et al., 2019) at a daily timestep and at a 5-arc minute (~10 km) spatial resolution applied to CONUS (section 2.1). In examining the interaction between hydrology and crop growth, we consider both one-way and two-way interactions. First, a one-way coupling is established to evaluate the effect of the simulated water availability of PCR-GLOBWB 2 for rainfed and irrigated crop growth in WOFOST (section 2.1; section 2.3.1). In addition, a two-way coupling is established in which, additional to passing water availability from PCR-GLOBWB 2 to WOFOST, the crop phenology of WOFOST in terms of actual evapotranspiration, leaf area index and rooting depth is fed back into PCR-GLOBWB 2 (section 2.1, 2.3.2). The justification for this coupling approach, along with technical implementation details, is elaborated upon in section 2.2.

Our framework was tested by comparing individual WOFOST and coupled one-way and two-way model runs to evaluate the impact of feedbacks on crop yield and irrigation water use (section 2.4). The results of these simulations are compared with and evaluated against reported yield statistics and reported annual irrigation withdrawals to assess their validity (section 2.5; section 3). In the end, we elaborate on the uncertainties, strengths, and usability of our coupled model framework for studying the water-food nexus under global change (section 4).

## 2 Methods

A newly coupled hydrological-crop model framework (Fig. 1) is developed to include the feedback between crop growth and hydrology. Here, we chose WOFOST as the crop growth model because of its detailed crop phenology and development and PCR-GLOBWB 2 as the hydrological model because of its detailed hydrological process simulation and large-scale applicability. This framework includes both a one-way and two-way coupling between the PCR-GLOBWB 2 global hydrological and water resources model (Sutanudjaja et al., 2018) and the WOFOST crop growth model (de Wit et al., 2019). The coupled framework was then used to quantify the impacts of included feedbacks between hydrology and crop growth on a daily timestep and 5 arcminutes resolution for CONUS.

The following (sub)sections provide a description of the PCR-GLOBWB 2 and WOFOST models and modules used (2.1), justification of coupling (2.2), the model coupling setup (2.3), model coupling simulation experiments and parametrization (2.3), validation of crop yield and of irrigation water use (2.4).

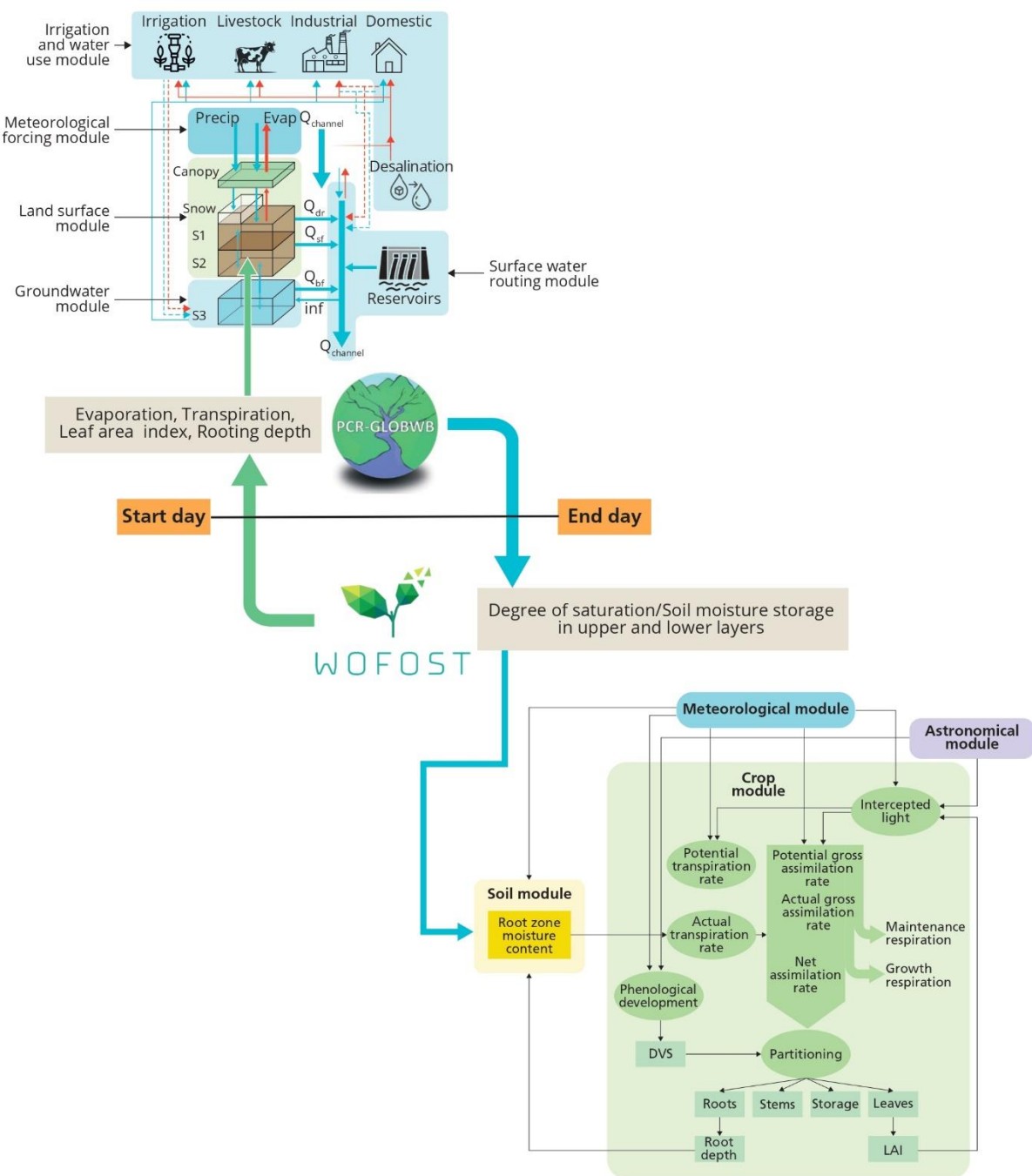

180

**Figure 1: The coupled model framework of the PCR-GLOBWB 2 hydrological and water resource model and the WOFOST crop growth model along with their model structures. The blue arrow represents the one-way coupling from PCR-GLOBWB 2 to WOFOST and the variables that are exchanged; the green arrow is added in case the full two-way coupling is considered. At the start of the day, WOFOST computes evapotranspiration, leaf area index, and rooting depth that is used by PCR-GLOBWB 2 to compute soil moisture status. At the end of the day, soil moisture storage in the upper and lower layers from PCR-GLOBWB 2 is fed to WOFOST to compute crop growth for the next day.**




**2.1. Model descriptions**
**PCR-GLOBWB 2**
The PCRaster Global Water Balance (PCR-GLOBWB 2) model (Sutanudjaja et al., 2018),
developed at Utrecht University, is a global hydrological and water resource model that
operates on a latitude-longitude grid. This model simulates the terrestrial hydrological cycle
with daily resolution, incorporating anthropogenic impacts like man-made reservoirs, sectoral
water demands, withdrawals, consumptive use, and return flows. PCR-GLOBWB 2 is applied
and tested across local to global scales.
PCR-GLOBWB 2 utilizes time-explicit schemes for all dynamic processes, running on daily
time steps for hydrology and water use, and sub-daily steps for hydrodynamic river routing. It
simulates moisture storage in two upper soil layers and manages water exchange among the
soil, atmosphere, and groundwater. Atmospheric interactions include precipitation,
evaporation, transpiration, and snow processes. The model considers sub-grid variability in
land use, soils, and topography, influencing run-off, interflow, groundwater recharge, and
capillary rise. Run-off is routed through river networks using methods ranging from simple
accumulation to kinematic wave routing, supporting floodplain inundation and surface water
temperature simulation.
The model includes a reservoir operation scheme for over 6000 human-made reservoirs from
the GRanD database, integrated according to their construction year. Human water use is
comprehensively modeled, estimating sectoral water demands and converting them into
withdrawals from groundwater, surface water, and desalination sources, while accounting for
resource availability and groundwater pumping capacity. Consumptive use and return flows
are calculated for each sector.
PCR-GLOBWB 2's flexible structure encompasses five main hydrological modules:
meteorological forcing, land surface, groundwater, surface water, irrigation, and water use. The
meteorological module uses gridded temperature and precipitation data. Reference potential
evaporation is calculated using Hamon's method and employed in the land surface module to
determine crop-specific potential evaporation. The groundwater and surface water modules
handle fluxes and stores for groundwater and surface water, respectively. The irrigation and
water use module simulates water demand, withdrawals, consumption, and return flows,
sourcing water from surface water (rivers and reservoirs), groundwater (both renewable and
non-renewable), and desalinated water, depending on availability. Detailed descriptions of
each module are provided by Sutanudjaja et al., (2018).
**WOFOST**
WOFOST (WOrld FOod STudies) is a crop simulation model developed at Wageningen
'School of De Wit', in the Netherlands, designed to quantitatively analyze the crop growth and
potential production of annual field crops at the field scale (Supit et al., 1994). WOFOST
employs a fixed time step of one day to simulate crop growth based on eco-physiological
processes such as phenological development and growth (de Wit et al., 2019). WOFOST has
found extensive application in assessing the impacts of climate change and management
strategies on crop growth and yield at local to global scales (Droppers et al. 2021).
The WOFOST crop model comprises of four modules: meteorological, crop, astronomical and
soil (Fig. 1). The WOFOST modules simulate a range of processes, including phenological
development, $CO_2$ assimilation, leaf development, light interception, transpiration, respiration,
root growth, assimilated partitioning to the various organs and the formation of dry matter. The
model's output includes simulated crop biomass total, crop yield and variables such as leaf area
and crop water use.
Temperature effects on crop development within WOFOST are modeled using temperature
sums, which accumulate daily temperatures above a specified threshold. These sums influence
germination and phenological stages, thereby affecting CO2 assimilation. Additionally, the
model accounts for the direct and indirect effects of suboptimal daytime temperatures on crop
growth and development, which are critical to overall plant performance. Daily photosynthesis
in the crop growth model is simulated by considering absorbed radiation and water stress. After
accounting for the assimilates used in maintenance respiration, the remaining resources are
allocated among the plant's leaves, stems, roots, and storage organs. A key internal driver of
this process is the leaf area index (LAI), which results from leaf area dynamics governed by
photosynthesis, biomass allocation, leaf age, and developmental stage. LAI, in turn, influences
the daily rates of photosynthesis.
WOFOST has been finely tuned to account for diverse climate and soil conditions, particularly
for commonly studied crops such as maize, soybean, and wheat, thereby, reducing the need for
further recalibration. This pre-tuning ensures that simulations reliably capture the growth and
yield responses of these crops under varying environmental conditions. For more detailed
information on the fine-tuning of crop variables, see (de Wit and Boogaard, 2021).

WOFOST employs a classic water balance approach designed for freely draining soils where groundwater is too deep to affect soil moisture content in the rooting zone. This approach divides the soil profile into two compartments: the rooted zone and the lower zone extending from the actual rooting depth to the maximum rooting depth. The subsoil below this maximum rooting depth is not considered. As roots extend deeper towards the maximum rooting depth, the lower zone gradually merges with the rooted zone. This approach is suitable for regional applications with limited soil property information. Soil moisture in the root zone serves as a primary link between the WOFOST model and the underlying soil module. For a detailed description of the WOFOST crop growth model, we refer to de Wit and Boogaard, (2021) and Supit et al., (1994).

## 2.2. Justification of model coupling

The integration of the hydrological model PCR-GLOBWB 2 (Sutanudjaja et al., 2018) with the crop growth model WOFOST (Supit et al., 1994) is crucial for accurately simulating the complex interactions between water availability and crop development. The hydrological model PCR-GLOBWB 2 is designed to simulate hydrological processes such as river discharge, groundwater flow, and water storage dynamics. It provides detailed representation and insights into the state and dynamics of water resources over large spatial scales and long temporal scales. On the other hand, the crop growth model WOFOST is focused on simulating crop phenology, including the stages of crop development, growth, and yield formation under varying environmental conditions.

Despite the strengths of each model, they individually have limitations that can affect the accuracy of simulations. PCR-GLOBWB 2 relies on static vegetation parameters, such as fixed Leaf Area Index (LAI) and root depth, which can limit its ability to reflect the dynamic nature of crop growth. On the other hand, WOFOST offers a detailed and dynamic representation of crop phenology and development, adjusting parameters like LAI and root depth based on actual growth stages. However, WOFOST employs a simplified water balance model, that may not adequately capture complex hydrological interactions.

To address these limitations, it is important to combine the strengths of both models to enhance hydrological and crop modelling performance. By integrating, WOFOST's detailed crop growth simulation capabilities with the robust hydrological process simulations of PCR-GLOBWB 2, we can better understand and represent the soil-plant-atmosphere interactions. Therefore, this study integrates PCR-GLOBWB 2 and WOFOST by passing soil moisture data

from PCR-GLOBWB 2 to WOFOST and feeding vegetative fluxes from WOFOST back into PCR-GLOBWB 2 on a daily basis. Additionally, to understand the intricate dynamics between hydrology and crop model, PCR-GLOBWB 2 is coupled to the WOFOST in one-way and two-way interactions.

In evaluating various coupling methods for integrating hydrological and crop models, we identified several approaches, including one where the hydrological model directly provides detailed irrigation schedules and percolation rates to the crop model. While this method offers highly detailed hydrological inputs, it often leads to inconsistencies due to the separate handling of soil moisture dynamics between the models, resulting in errors in soil moisture management and water balance. Commonly used coupling procedures, such as those described by Li et al., (2014) and Tsarouchi et al., (2014), calculate potential evapotranspiration and vegetation water uptake within the hydrological model, which is then passed to the crop model to simulate crop growth. The crop model then calculates state variables like leaf area index, root depth, and canopy height, which are subsequently fed back into the hydrological model. However, these methods can introduce system errors, particularly in the transpiration module, if there is a discrepancy between evapotranspiration calculated by the crop and hydrological model, as highlighted by Wang et al., (2012). Our chosen coupling method, where soil moisture is calculated by PCR-GLOBWB 2 and passed to WOFOST and vegetative dynamics and evapotranspiration fluxes are then fed back into PCR-GLOBWB 2, offers a balanced approach that ensures consistency, and the necessary complexity, and efficiency in the simulations.

The selected coupling approach also addresses specific challenges associated with the models. PCR-GLOBWB 2 allows for flexible land cover classification and parameterization, which is essential for accurately representing diverse crop types and their interactions with water resources. For this study, we defined 12 land cover types (tall natural, short natural, pasture, irrigated maize, irrigated soybean, irrigated wheat, non-paddy irrigated crops (irrigated other crops), paddy irrigated crop, rainfed maize, rainfed soybean, rainfed wheat and rainfed others. WOFOST's role in this coupling is to pass the fluxes of irrigated and rainfed maize, soybean, and wheat to PCR-GLOBWB 2, ensuring a detailed simulation of crop water use.

One of the key considerations in this coupling is accurately calculating the soil-water balance. Given its more advanced soil moisture accounting scheme, PCR-GLOBWB 2 handles this aspect, as WOFOST's simpler single-layer leaky bucket approach could introduce complexities if soil moisture data were passed from WOFOST to the multi-layered soil model

of PCR-GLOBWB 2. Therefore, the coupling approach we selected minimizes potential
discrepancies while maximizing the strengths of each model.
It is important to acknowledge, that individual models come with inherent uncertainties, related
to model structure, parameters, and data. When coupling these models, the level of uncertainty
compounds further (Kanda et al., 2018). Additionally, the nature of coupling itself can
introduce another layer of uncertainty. According to Antle et al., (2001), coupling models lead
to further conceptualization and computational problems, elevating uncertainty levels.
Therefore, an efficient coupling is essential to minimize these risks. There are three primary
methods for coupling models (Vereecken et al., 2016): light/loose coupling,
external/framework coupling using a central coupler, and full coupling.
In light or loose coupling, the output of one model serves as the input for the other, which can
lead to a straightforward but limited interaction. Framework coupling uses a central coupler for
communication between models without requiring code modification, offering a balance
between integration and flexibility. Full coupling involves both models sharing the same
boundary conditions, drivers, and variables, which requires significant code modification.
**Implementation of the (BMI) framework coupling**
Given the complexity of integrating the PCR-GLOBWB 2 and WOFOST models and the need
for efficient simulations, we opted for framework coupling. This approach was chosen because
WOFOST and PCR-GLOBWB 2 are written in different programming languages (C and
PCRaster-Python, respectively). Framework coupling allows for seamless interaction between
the models at each time step, facilitating dynamic exchanges while limiting I/O-related
computation times. We employed the Basic Model Interface (BMI) for this purpose (Hutton et
al., 2020; Peckham et al., 2013). The decision to use BMI over alternative techniques was
driven by its non-interfering nature, ensuring no code entanglement and facilitating seamless
connection between the two models. BMI functions act as a bridge, enabling direct variable
exchange between WOFOST and PCR-GLOBWB 2 without modifying their source code. This
non-invasive approach ensures a flexible and robust coupling framework, allowing continuous
model development without interruptions. Integrating BMI functions into both models
provides a set of functions for retrieving or altering model variables, enhancing adaptability,
and efficiency.
An additional wrapper was required to translate the model-specific BMI functions into Python-
compatible information to establish a Python-based coupling framework. The Babelizer
wrapper (CSDMS, 2024) was utilized for this purpose with the WOFOST BMI. Conversely,
no supplementary wrapper is needed in the PCR-GLOBWB 2 BMI, as the model is inherently
Python-compatible due to its programming language.
The Babelizer wrapper facilitates the integration of the WOFOST model by utilizing an input
file that provides essential details, including the model library, entry point, packages, and
author information. This input file guides the construction of the necessary dependencies to
generate Python bindings. Once these Python bindings are created, Babelizer ensures the
successful integration of the WOFOST BMI into Python by verifying that the bindings are
correctly built and loaded.

**359 Workflow of PCR-GLOBWB 2 - WOFOST model framework**

In the PCR-GLOBWB 2 - WOFOST coupling framework, the workflow after implementing
BMI functions remains consistent for both one-way and two-way coupling, up until the
initialization of the hydrological and crop models (Fig. 2).

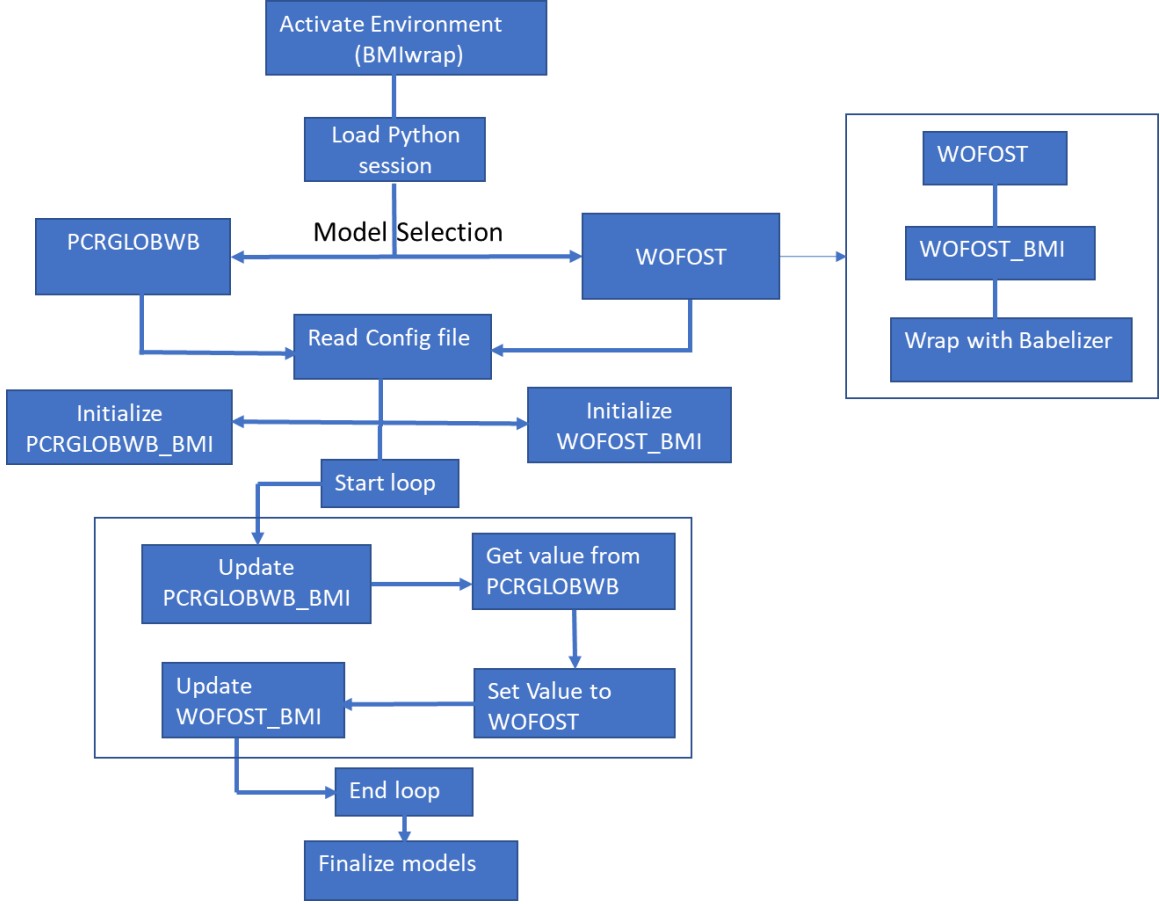


**Figure 2: Schematization of the workflow of the coupled PCR-GLOBWB 2 - WOFOST model framework**
Before initiating the Python session, it is crucial to activate the BMI wrap environment, which
includes all necessary libraries for both hydrological and crop models. After this setup, the
PCR-GLOBWB 2 and WOFOST models, along with their configuration files that define the
coupling settings, are loaded into the Python session. BMIwrap reads the configuration file,
initializing the model-specific configuration settings before establishing both models as a
coupled entity.
Once the coupled models are initialized, a loop is initiated, commencing at the start time and
concluding at the end time. During each iteration of this loop, variables are exchanged between
the models based on the one-way or two-way coupling configuration. This iterative process
ensures a continuous and seamless flow of information between the PCR-GLOBWB 2
hydrological model and the WOFOST crop model throughout the simulation period.
**2.3.Model coupling setup**
The developed PCR-GLOBWB 2 - WOFOST coupled model framework integrates
hydrological and crop models through both one-way and two-way couplings, as illustrated in
Fig. 1 &3. This model coupling aims to assess the intricate interactions between hydrology and
crop growth under different agricultural conditions, specifically irrigated and rainfed settings.
The one-way coupling examines the impact of water availability on crop growth, while the
two-way coupling incorporates the exchange of soil moisture status and hydrological
parameters and fluxes based on crop status.

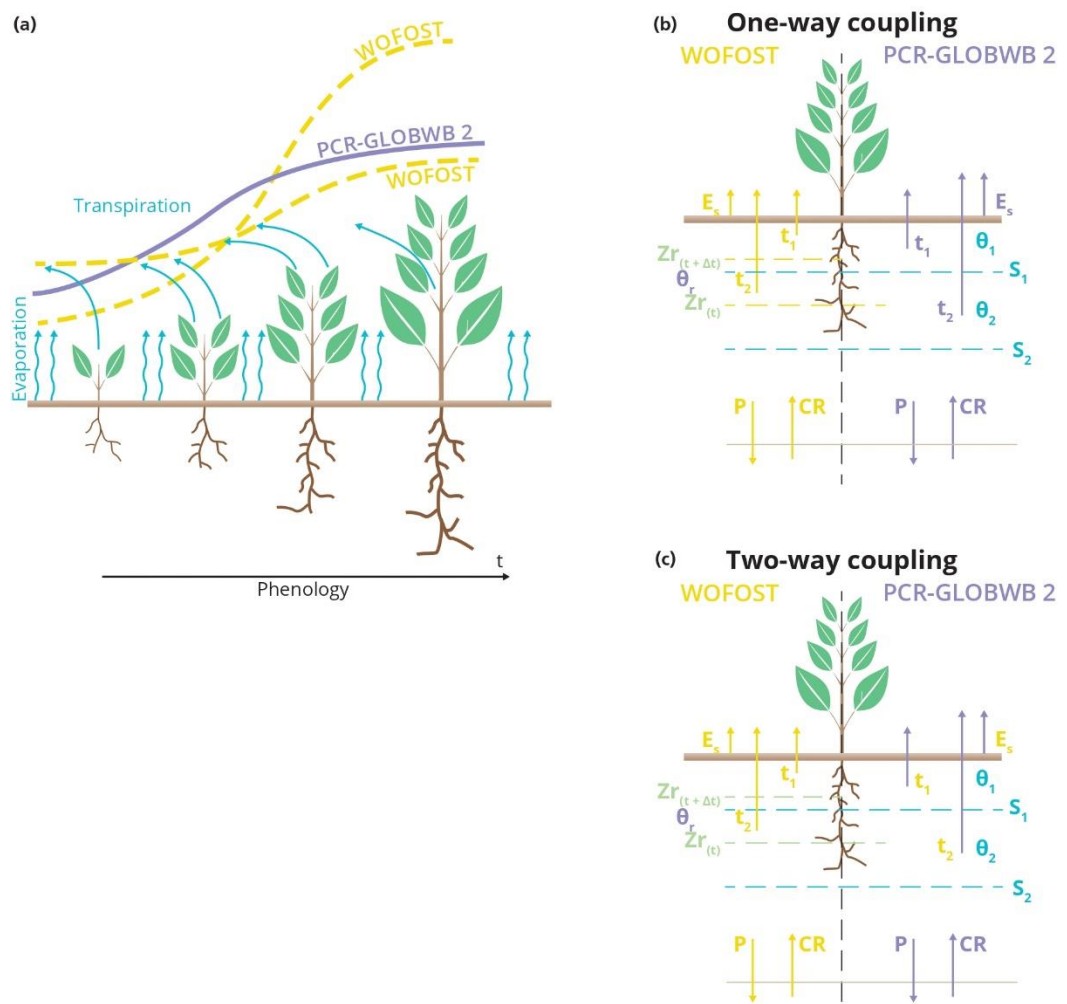


**Figure 3: Schematic view of the coupled model framework: a) shows the calculated phenology from WOFOST and PCR-GLOBWB 2 over time along with the associated fluxes. b) displays a detailed representation of the one-way coupling approach, where soil moisture is transferred from PCR-GLOBWB 2 to WOFOST and (c) illustrates the two-way coupling approach, where variables are exchanged in both directions between PCR-GLOBWB 2 and WOFOST.**

### 2.3.1. One-way coupling

In the one-way coupling, information on soil moisture status is passed from PCR-GLOBWB 2 to WOFOST (Fig 3(b)). Here, PCR-GLOBWB 2 simulates soil moisture content for every day and the soil water storage is simulated separately for each land cover type. Consequently, WOFOST receives the soil moisture content from PCR-GLOBWB 2 as input, with generally higher values of soil moisture for irrigated crops than of nearby rainfed crops. WOFOST then simulates the crop yield based on the simulated soil moisture content and the same meteorological inputs as PCR-GLOBWB 2 uses.

The combined model framework captures the impact of hydroclimatic conditions by assessing water stress and heat stress. Water stress, influenced by soil moisture levels derived from PCR-

GLOBWB 2, affects various processes in WOFOST such as a reduction in the leaf area, a decrease in the assimilation of biomass (growth), changes in the partitioning of biomass, and an increase in various plant organs of senescence (ageing processes). Elevated temperatures have varying effects across different stages of crop development. They can accelerate crop growth by promoting faster accumulation of Growing Degree Days, which are essential for determining crop maturity. However, prolonged exposure to high temperatures can also induce heat stress, adversely impacting crop health and potentially shortening the overall duration of the crop's growth cycle. Insufficient water availability that limits the evapotranspiration also reduces the amount of assimilation and the corresponding yield.

### 2.3.2. Two-way coupling

- In addition to one-way coupling, the two-way coupling approach involves iterating data exchange between WOFOST and PCR-GLOBWB 2 twice per day. WOFOST calculates the vegetation states, (such as leaf area index (LAI), biomass, and root depth) and fluxes (e.g., evapotranspiration) for irrigated and rainfed maize, soybean, and wheat crops, while other vegetation and non-vegetation fluxes for other crops are simulated within PCR-GLOBWB 2. To be more specific, for the fraction of land cover that is different from maize, wheat, and soybean, the vegetation states and fluxes are calculated within the PCR-GLOBWB 2. For these land cover types, vegetation phenology in the form of crop factors, is approximated by a yearly climatology. In the two-way coupling, data is exchanged between PCR-GLOBWB 2 and WOFOST as follows (Fig. 3c): At the start of the day, PCR-GLOBWB 2 passes the previous day's soil moisture to the WOFOST, assuming no root development has occurred overnight. WOFOST then computes the potential evapotranspiration based on the meteorological variables at the current time step and the pertinent vegetation states from the previous time step (leaf area index (LAI), rooting depth, and crop height). It also calculates the actual bare soil evaporation, actual transpiration (actual evapotranspiration), potential evaporation, and open water evaporation;

- The calculated fluxes are passed to PCR-GLOBWB 2, together with the root depth. The root depth is used to partition the actual transpiration from the single root zone of WOFOST over the two soil layers of PCR-GLOBWB 2, dependent on the root content. For both irrigated and rainfed crops, the actual evapotranspiration from WOFOST is forced to PCR-GLOBWB 2 and used to update the soil moisture content of the two soil layers in PCR-GLOBWB 2 for the current daily timestep;

- In the case of irrigated crops, the stages of vegetated development are used to compute the amount of irrigation in PCR-GLOBWB 2. Potential evaporation is used to calculate the irrigation water demand for paddy crops (not considered here), whereas the irrigation water requirement for non-paddy crops is computed based on the soil moisture status according to the FAO guidelines (Allen et al., 1998). The irrigation water requirement is withdrawn from the available water resources in PCR-GLOBWB 2, and the available irrigation water supply is applied to the crops in addition to any natural precipitation;

- At the end of the day, the resulting soil moisture from the two soil layers from PCR-GLOBWB 2 is aggregated to provide a total for the root zone of each crop, which is then passed back to WOFOST;

- Using the updated soil moisture from PCR-GLOBWB 2, WOFOST computes the actual transpiration and updates crop growth and the crop status. The new fluxes and crop parameters are then passed to PCR-GLOBWB 2 again on the next day (Fig.1, Fig. 3c).

In this two-way coupling, the crop phenology from WOFOST determines evapotranspiration and thus the soil hydrology of PCR-GLOBWB 2, particularly during dry spells. Compared to the predefined phenology of PCR-GLOBWB 2, the LAI, rooting depth and evapotranspiration as simulated by WOFOST will lag during dry spells and less water may be lost from PCR-GLOBWB 2. However, the thinner rooting depth will also lead to an earlier drying out of the soil and reduced capillary rise. This subsequently leads to reduced soil moisture (compared to PCR-GLOBWB 2 stand-alone) which in turn feeds back to a reduced simulated yield in WOFOST, in particular for rainfed crops. For irrigated crops, the extra water supplied will largely offset these feedbacks and result in near-optimum growth.

## 2.4. Model coupling simulation experiments and parametrization

Hydrological simulations were conducted with a daily timestep at a 5-arcminute grid resolution, where for each grid cell WOFOST was used to simulate crop growth for irrigated and rainfed maize, soybean, and wheat. To assess the impact of hydrology on crop growth and understand the interactions between hydrology and crop growth, three sets of simulations were carried out for both irrigated and rainfed crops: a) stand-alone simulations using the WOFOST crop model solely, b) one-way coupled, and c) two-way coupled PCR-GLOBWB 2 – WOFOST simulations. Note that for the stand-alone simulations with WOFOST under irrigation the potential crop yield is simulated, which is potential yield without water (and nutrient) stress

except for temperature effects. When coupled to PCR-GLOBWB 2, water stress can occur even
for irrigated crops in case there is not enough water available (in PCR-GLOBWB 2) to fully
satisfy the crop water demand. For rainfed crops, growth is influenced by available soil
moisture for all simulations and is thus sensitive to water stress and temperature. Green water
from natural rainfall is the primary water supply in rainfed analysis, while irrigated crops get
water from both green and blue water (from surface water and renewable groundwater) and
non-renewable groundwater leading to groundwater depletion.
Daily timestep simulations covered the period from 1979 and 2019, using weather variables
(minimum and maximum air temperature, short wave radiation, precipitation, vapour pressure,
windspeed, and humidity) from the W5E5 forcing data (Lange et al., 2021) as input to PCR-
GLOBWB 2 (Sutanudjaja et al., 2018) and WOFOST. Cropland areas and growing seasons
were determined from the MIRCA2000 (Portmann et al., 2010) global monthly irrigated and
rainfed crop area dataset. The focus of the coupled framework was to comprehend the impacts
and feedback between hydrology and crop growth. Crop parameters, atmospheric $CO_2$
concentrations, and fertilizer application were obtained from the WOFOST crop parameter
dataset for each crop (WOFOST Crop Parameters, 2024). Cultivars in the WOFOST crop
parameter datasets were calibrated for each crop against reported agricultural yields from the
United States Department of Agriculture (USDA) National Agricultural Statistics Service
(USDA, 2024), with the closest matching cultivar selected for final simulations. Detailed
information on the cultivar calibration for each crop (i.e., irrigated and rainfed maize, soybean,
and wheat) is provided in the supplementary information section II.
Additionally, to ensure a consistent comparison, we harmonized the soil parameters in both
WOFOST and PCR-GLOBWB 2 by incorporating data from the FAO soil map (FAO, 2007).
WOFOST uses constant soil parameters across all spatial locations, which may not accurately
represent local soil variability. By integrating FAO soil data, we ensured consistency in soil
properties such as water-holding capacity and infiltration rates across the different models,
improving the robustness of the comparison.
Comparisons were made between simulations from stand-alone WOFOST and the one-way
and two-way coupled PCR-GLOBWB 2 - WOFOST runs. This comparative analysis involved
evaluating the results from different model runs for crop growth against reported crop yields.
Furthermore, irrigation water withdrawals of coupled model runs are compared against the
USGS Water Use Database (USGS, 2023) (section 2.4).

## 2.5. Model evaluation

We evaluated the three different model configurations by comparing simulated results against reported USDA crop yields of maize, soybean, and wheat. Furthermore, we cross-referenced our simulations with irrigation water withdrawal data spanning five years from the USGS Water Use Database. Specifically, we compared data for the years 2005, 2010, and 2015, as the USGS census data is collected at five-yearly intervals.

### 2.5.1. Crop yields model evaluation

To assess the model's performance, we employ three key metrics: correlation coefficients (r), Normalized Root Mean Square Error (NRMSE), and Normalized Bias (NBIAS). These metrics were selected for their ability to capture the strength, accuracy and systematic errors in the relationship between simulated and observed values.

$$r = \frac{\sum (P_i - \bar{P})(O_i - \bar{O})}{\sqrt{\sum (P_i - \bar{P})^2 \cdot \sum (O_i - \bar{O})^2}} \tag{1}$$

$$NRMSE = \frac{\sqrt{\frac{1}{n} \sum_{i=1}^{n} (P_i - O_i)^2}}{\bar{O}} \tag{2}$$

$$NBIAS = \frac{\frac{1}{n} \sum_{i=1}^{n} (P_i - O_i)^2}{\bar{O}} \tag{3}$$

Where, $P_i$ and $O_i$ are the individual predicted and observed values, respectively and $\bar{P}$ and $\bar{O}$ are the means of the predicted and observed values.

The evaluation was done both temporally for average CONUS yields per year, as well as for multi-year averages per state-per-state to evaluate the model's ability to capture spatial variations in crop yield. This was done for both irrigated and rainfed maize, soybean, and wheat.

To further characterize the dataset and evaluate the impact of the degree of coupling on simulated yields, additional statistical analyses were conducted on the 41 years of simulated data at the 5-arcminute grid scale. To this end, the mean and coefficient of variation (CV) were computed for both one-way and two-way datasets for the three crops under irrigated and rainfed conditions. The purpose of this analysis was to examine the central tendency and year-to-year variability of yield simulations and how these are related to the way hydrology and crop growth are coupled.

### 2.5.2. Irrigation water use model evaluation

The USGS reported irrigation water use data provides a comprehensive representation of the total irrigation water utilized by all crops for a number of states (USGS, 2023). The irrigated crop area used in this dataset is however not the same as that used in PCR-GLOBWB 2, which is based on MIRCA2000 (Portmann et al., 2010). Thus, directly comparing USGS data with our simulated water withdrawals would result in bias. To ensure a fair comparison between the simulated and reported data for all crops, we adjusted the USGS irrigation water use data by multiplying these with the ratio of the irrigated area from MIRCA2000 to the reported total USGS irrigated area. Additionally, our simulated irrigation water withdrawal volumes did not yet account for irrigation efficiency. We intend to implement this in future development. Hence, we introduced an additional correction by dividing the simulated withdrawal data by the irrigation efficiency as is commonly used in PCR-GLOBWB 2 when it is not coupled to a crop model.

After these corrections, the coupled model simulated irrigation water withdrawals for all crops were evaluated against actual irrigation data obtained from the USGS database through spatial (multi-year averages per state) and temporal (multi-state totals per year) analysis, providing insights into the model's ability to replicate observed irrigation water use patterns.

This comparison was limited to the years with available reported area data for the simulation period (2005, 2010, 2015) and to the states with reported irrigation water withdrawal volumes for these years (37 states).

**3. Results**

In this section, we present the key findings obtained from the implementation of the coupled hydrological-crop growth model framework based on WOFOST and PCR-GLOBWB 2. We present our findings sequentially, first delving into observed hydrological impacts on crop growth (one-way coupling) and then exploring how feedback mechanisms between crop growth and hydrology impact the crop growth system (two-way coupling).

**3.1 Comparative temporal and spatial analysis of stand-alone, one-way, and two-way coupling for irrigated and rainfed crops**

Temporal analysis (Fig. 2) compares the simulated yields with reported yields for irrigated and rainfed maize, soybean, and wheat crops spanning from 1979 to 2019 in the CONUS region. Notably, the reported yields exhibit discernible trends for the CONUS region across the three crops and in both irrigated and rainfed analysis. This temporal evolution is primarily attributed to technological advancements, encompassing improved agricultural practices and the

introduction of enhanced crop varieties over the study period (Arata et al., 2020). In contrast,
simulated yields of our coupled PCR-GLOBWB 2–WOFOST model framework do not capture
such trends, as the modelling approach intentionally omitted to incorporate trends in
technology and management practices. This intentional omission was to focus on the intrinsic
biophysical processes and climatic conditions affecting crop yields, providing a baseline
understanding unaffected by external advancements.
The trends in reported yields differ significantly across all crops and between irrigated and
rainfed systems. For maize, both irrigated and rainfed yields show an increasing trend,
particularly post-2000, which is not reflected in the simulated yields. Soybean yields exhibit a
gradual upward trend in irrigated systems, while rainfed soybean yields show little to no
discernible trend until 2007, followed by a slight increase. Wheat yields, both irrigated and
rainfed, demonstrate fluctuations with a slight upward trend towards the end of the period.
These discrepancies can be attributed to various factors, including technological advancements,
improved agricultural practices, and the introduction of enhanced crop varieties, which were
not incorporated into the modelling approach. To ensure a consistent and meaningful analysis,
we selected the years 2006-2019 for further analysis (spatial analysis (Fig. 5) and evaluation
metrics (Table. 1)). This period was selected because reported yields during these years appear
more stable and are better aligned with the simulated yields, allowing for a fair evaluation of
the model's accuracy and reliability. For the selected periods, we think that the results are
convincing, and, except for rainfed Soybean, they are certainly up to par with the results from
other crop growth modelling studies at continental scales.

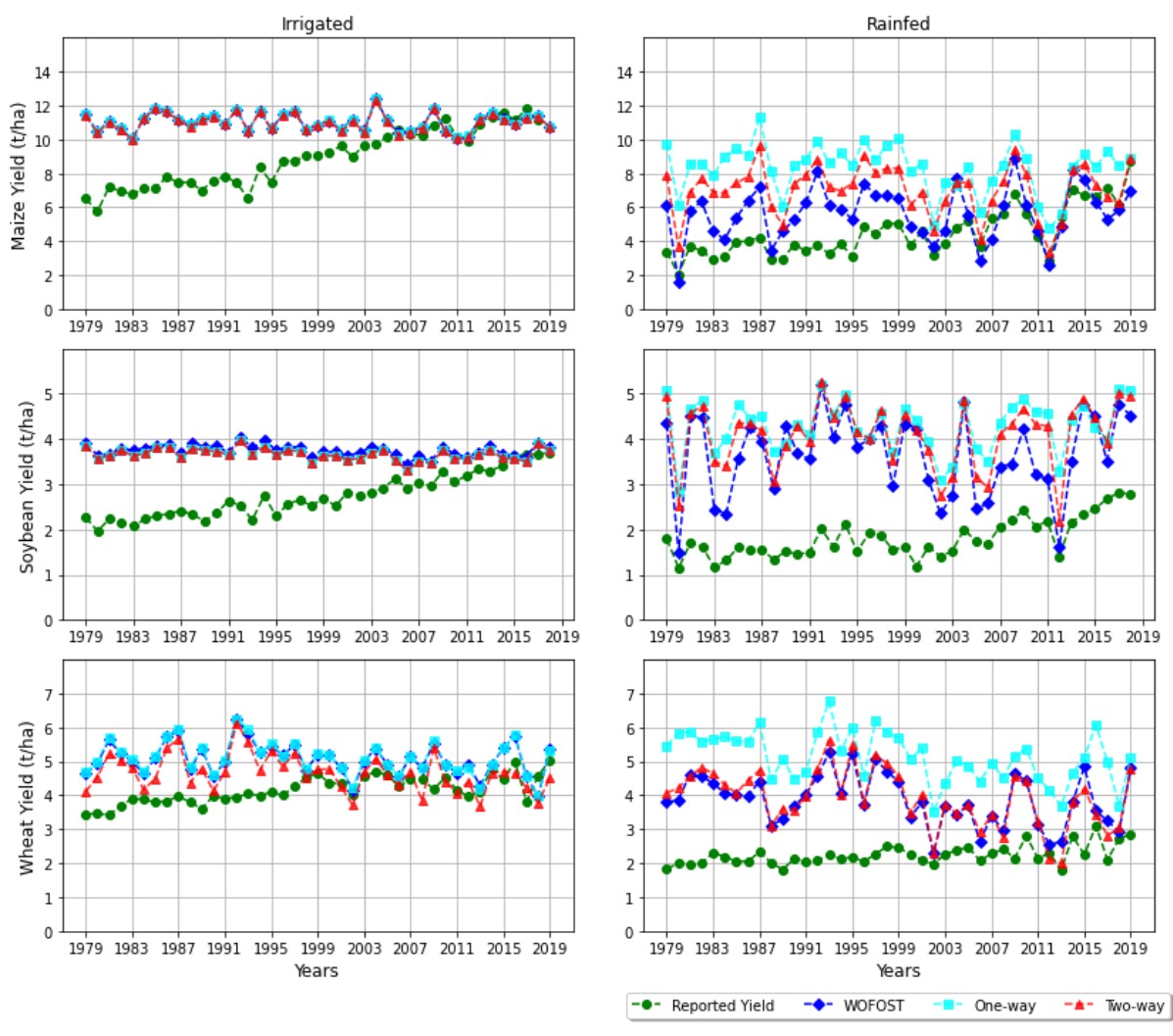


**Figure 4: Temporal analysis of irrigated and rainfed crops of a) maize, b) soybean and c) wheat for the years 1979 to 2019 of a CONUS region**


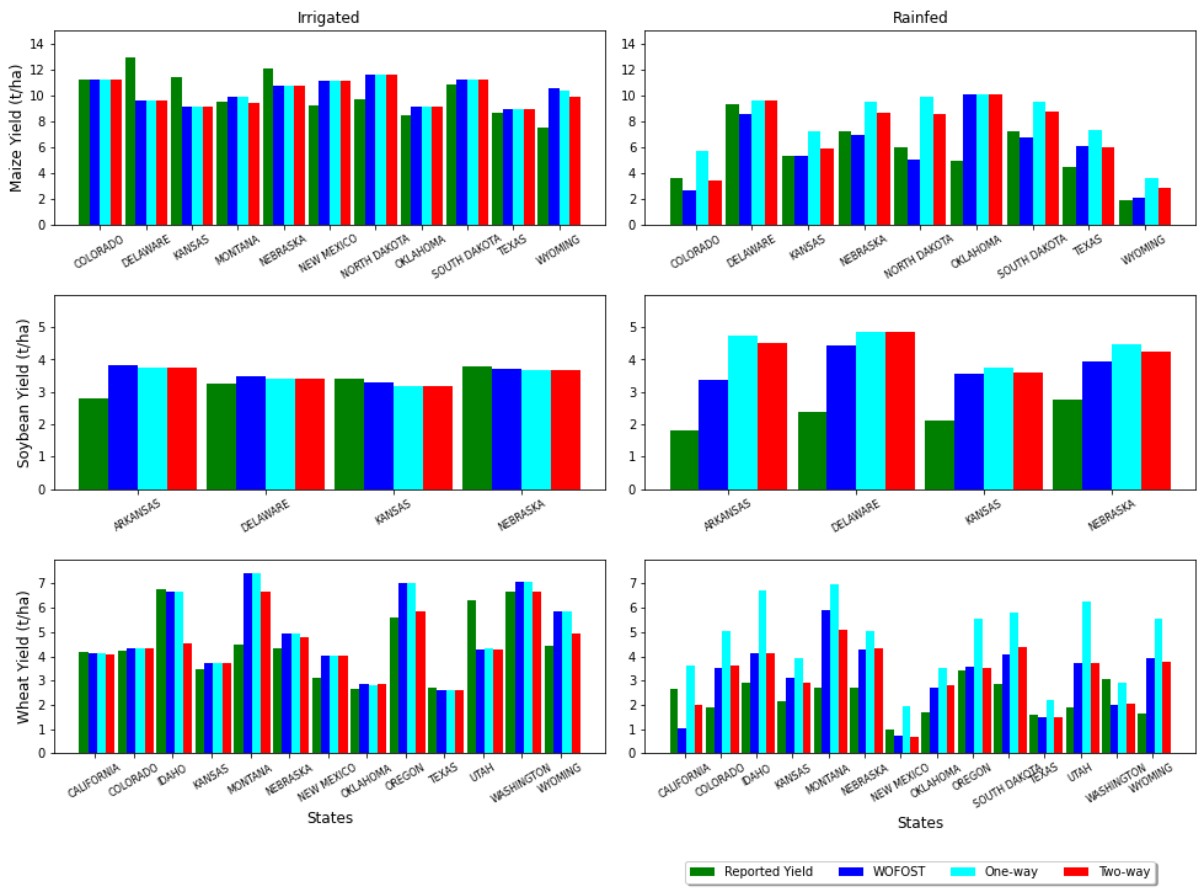

**Figure 5: Spatial (i.e., state level) analysis of irrigated and rainfed crops of a) maize, b) soybean and c) wheat for the years 2006 to 2019 for the CONUS region.**

Figures 4 and 5 show the outcomes of comparing simulated irrigated and rainfed analyses yields for maize, soybean, and wheat with reported yields. For the irrigated crops, the obtained yields by stand-alone WOFOST represent the potential productivity for the three crops. Notably, one-way and two-way model runs for irrigated crops yielded nearly identical results to the stand-alone runs, indicating that there is generally enough irrigation water to completely satisfy crop water demands. This similarity arises because in irrigated conditions, water supply is managed to meet crop water demands fully, thereby minimizing the influence of soil moisture variability on yield outcomes. In other words, since the primary constraint, water availability, is alleviated by irrigation, the simulations naturally converge, regardless of the model coupling approach. Although not shown here, we note that this is at the expense of non-renewable groundwater use in states overlying the Southern Great Plains aquifer system.

Conversely, for rainfed crops that rely solely on rainfall, we generally expect similar yields from stand-alone and two-way coupled simulations since the primary water input is rainfall. However, differences were observed between these models, more pronounced in maize crops and less significant in soybean and wheat, with yields in the two-way coupled model being

larger than stand-alone WOFOST. These differences can be attributed to various factors. The coupled model incorporates detailed soil moisture dynamics, including processes like percolation, capillary rise, and surface runoff, which directly influence water availability for crops. For example, higher capillary rise from groundwater can increase soil moisture, thereby increasing water available to crops, whereas surface runoff limits infiltration, and deep percolation rates lead to water loss beyond the root zone, reducing available moisture. In contrast, stand-alone WOFOST cannot accurately capture such variability, leading to differences in simulated yields.

Another key distinction lies in how the plants access soil moisture in the root zone in the different models. In the stand-alone WOFOST model, all soil moisture is extracted from a single soil layer using a simple one-layer tipping bucket approach. Conversely, PCR-GLOBWB 2 subdivides the soil profile into two layers (see section 2.1), with evapotranspiration distributed between them. In this setup, bare soil evaporation is entirely sourced from the upper layer, while transpiration is drawn from both layers. In the coupled approach, soil moisture can be disproportionately supplied from the wetter second layer, including the part of the second soil layer that is below the root zone. This provides slightly higher average root zone soil moisture to WOFOST under stressed conditions, making the simulated yields of the two-way-coupled PCR-GLOBWB 2- WOFOST a bit higher than those of stand-alone WOFOST  (see Supplementary Information III-2, Supplementary Figure S6-S9 for a detailed analysis of the differences between stand-alone WOFOST, one-way and two-way coupled PCR-GLOBWB 2 – WOFOST models).

The one-way coupling approach generally overestimates yields relative to stand-alone and two-way simulations, particularly for wheat and, to a lesser degree, for maize. This discrepancy arises from the fact that in one-way coupling, the phenology (Leaf area index and root development) is prescribed and independent of actual crop development as simulated by WOFOST. In a dry and warm year, crop development in WOFOST is faster than average and thus also faster than the fixed development in PCR-GLOBWB 2; this is due to higher radiation and temperature at the beginning of the growing season. This leads to early-season higher evapotranspiration in the stand-alone WOFOST model and less available soil moisture at the end of the season when the storage organs are formed (see Supplementary Information III-1). In a one-way coupled setup, plant development from WOFOST is not fed back to PCR-GLOBWB 2, so that soil moisture in PCR-GLOBWB 2 remains higher throughout the season. This higher soil moisture is passed to WOFOST, leading to higher yields in the one-way

coupled PCR-GLOBWB 2 – WOFOST model than the stand-alone WOFOST model and the
two-way coupled PCR-GLOBWB 2 – WOFOST model, where crop development is fed back
to PCR-GLOBWB 2.  We further refer to Supplementary Information III, Supplementary
Figure S4-S9, for a detailed analysis of the differences between stand-alone WOFOST, one-
way and two-way coupled PCR-GLOBWB 2 – WOFOST models.
The temporal analysis (Fig. 4) of simulated and reported yields reveals distinct trends and year-
to-year fluctuations for each crop. For maize, both irrigated and rainfed conditions show
considerable variability in yields over the years. Rainfed maize, in particular, exhibits a
discernible pattern with certain years marked by notable peaks in yields, highlighting its
sensitivity to varying environmental conditions. These variations are also observed in reported
maize yields. This indicates that maize yields, especially under rainfed conditions, are highly
influenced by annual climatic variability.
For wheat, the simulated yields under both irrigated and rainfed conditions show similar year-
to-year patterns, which are not as evident in the reported yields. This suggests that the
discrepancies might be due to the model's sensitivity to water and temperature variability,
which may not fully capture the complexities of actual wheat production. Specifically, factors
such as the use of different wheat varieties, the differentiation between winter and spring wheat,
and their respective growth parameters could influence the observed yields. These varietal and
seasonal distinctions introduce variability that the model might not fully incorporate, leading
to differences between simulated and reported yields.
Soybean yields present a different scenario. Both irrigated and rainfed simulated yields
consistently surpass the reported values, with the discrepancy being more pronounced in
rainfed conditions. This overestimation could be due to the model's assumptions or parameters
that do not fully capture the limitations faced by soybean crops in real-world rainfed
environments, such as variations in soil fertility, pest pressures, crop varieties and other
management practices not accounted for in the model.
In the spatial analysis (Fig. 5), simulated irrigated maize yields from stand-alone (WOFOST),
one-way, and two-way coupling align almost identical with reported irrigated maize yields.
Conversely, in rainfed maize analysis, both stand-alone and two-way simulations closely
matched reported yields in states such as Delaware, Colorado, Kansas, Nebraska, South Dakota
and Wyoming, while one-way coupling exhibits an overestimation of yields compared to stand-
alone (WOFOST) and two-way coupling.

For soybeans, the spatial analysis reveals identical yields among stand-alone (WOFOST), one-way, and two-way simulations for both irrigated and rainfed crops. For irrigated crops, simulated yields were overestimated in Arkansas state but closely matched in states like Delaware, Kansas, and Nebraska compared to reported values. Under rainfed conditions, all three models overestimated the simulated yield relative to reported yields. For irrigated wheat, simulated yields of the two-way coupling outperform stand-alone WOFOST and one-way coupling, particularly in states like Oregon, Washington, and Wyoming. In contrast, for rainfed wheat, stand-alone and two-way coupling simulations closely align except in states such as California and Montana. The one-way coupling, lacking feedback from the crop growth model to the hydrological model, leads to an overestimation of rainfed yields across all states compared to stand-alone WOFOST and two-way coupling. This underscores the importance of incorporating two-way interactions and feedback mechanisms for more accurate yield simulation results.

**3.2 Evaluation statistics**

Table 1 presents model performance metrics (correlation, normalized RMSE and normalized bias) from the temporal analysis, evaluating simulations for the three model setups (i.e., stand-alone WOFOST, one-way, two-way coupling) for irrigated and rainfed maize, soybean, and wheat for the period 2006-2019 (see section 3.1). Model performance metrics for the spatial analysis are presented in supplementary IV.

For irrigated crops, simulations of all model approaches exhibit positive correlations with reported yields, though correlation coefficients vary across models and crops. Two-way coupling shows a slightly higher correlation (0.59) with reported yields for maize, but a lower correlation (0.24) for wheat compared to stand-alone and one-way coupling. The root mean square errors (RMSE) normalized to the mean remain consistently low, with values ranging from 0.08 to 0.15 across three crops, indicating a reasonable fit of the simulated values to the observed data. Moreover, normalized biases are also low, ranging from -0.65 to 0.50. The two-way coupling demonstrates overall slightly lower biases and minimal error compared to stand-alone and one-way simulations, particularly for wheat.

**Table 1: Model performance metrics (i.e. correlation, normalized RMSE and normalized bias) for simulated irrigated and rainfed maize, soybean, and wheat.**

| S.NO | Metrics | Maize | | | Soybean | | | Wheat | | |
|------|---------|-------|---|---|---------|---|---|-------|---|---|
| **Irrigated crops** | | **Stand alone** | **One-way** | **Two-way** | **Stand alone** | **One-way** | **Two-way** | **Stand alone** | **One-way** | **Two-way** |
| 1 | Correlation | 0.57 | 0.56 | 0.59 | 0.25 | 0.36 | 0.36 | 0.45 | 0.46 | 0.24 |
| 2 | Normalized RMSE | 0.08 | 0.08 | 0.08 | 0.12 | 0.10 | 0.10 | 0.15 | 0.15 | 0.11 |
| 3 | Normalized Bias | -0.58 | -0.58 | -0.65 | 0.34 | 0.27 | 0.27 | 0.50 | 0.50 | -0.02 |
| **Rainfed crops** | | | | | | | | | | |
| 1 | Correlation | 0.80 | 0.81 | 0.84 | 0.86 | 0.80 | 0.84 | 0.37 | 0.46 | 0.47 |
| 2 | Normalized RMSE | 0.19 | 0.50 | 0.20 | 0.68 | 1.01 | 0.91 | 0.56 | 0.97 | 0.55 |
| 3 | Normalized Bias | -0.10 | 2.00 | 0.91 | 1.39 | 2.24 | 2.03 | 1.13 | 2.26 | 0.98 |

For rainfed crops, the correlation coefficients vary, with two-way coupling displaying the highest correlations. Higher correlation coefficients are obtained for maize (0.808-0.84) and soybean (0.84-0.86) compared to wheat (0.37-0.47). Normalized RMSE values are generally higher in rainfed conditions compared to irrigated, ranging from 0.19 to 1.01. Normalized biases show variations across simulation approaches and crops, ranging from -0.10 to 2.26. Specifically, one-way coupling exhibits higher biases in rainfed maize, soybean, and wheat compared to stand-alone and two-way simulations. Two-way coupling shows lower error in wheat crops compared to the stand-alone model, while the stand-alone model performs better for maize and soybeans than both two-way and one-way coupling.

Overall, the validation results affirm the overall effectiveness of the simulation approaches in accurately representing observed irrigated and rainfed crop yields, with stand-alone and two-way coupling slightly outperforming one-way simulations.

### 3.3 Relevant feedbacks revealed by two-way coupling between hydrology and crop growth

We further investigated the impact of the developed model coupling by looking at its impact on simulated crop yield in terms of the CONUS-wide 5-arcminute spatial variation and multi-year variability. To evaluate the impact of coupling dynamics, we assessed key indicators, including mean crop yields, the coefficient of variation (CV) of crop yields expressing

interannual variability, and the relative difference in mean and CV between two-way and one-
way couplings.
Spatial patterns of the 1979-2019 mean simulated crop yields of maize, soybean and wheat are
shown under irrigated (Fig. 6) and rainfed (Fig. 7) conditions across the CONUS region. The
stand-alone simulations show the yield distribution without coupling between the hydrological
and crop models, relying on the internal soil moisture calculation using a simple one-layer
tipping bucket approach. In contrast, one-way and two-way coupled simulations involve
dynamic interaction between the hydrological model (PCR-GLOBWB 2) and crop growth
model (WOFOST), where soil moisture from PCR-GLOBWB 2 is passed to the WOFOST,
with two-way coupling also incorporating feedback from WOFOST to the PCR-GLOBWB 2
(see Fig 3).
For irrigated crops (Fig. 6), the regions show similar yields for stand-alone, one-way and two-
way coupled simulations. This is expected since soil moisture is kept at optimal levels in
irrigated conditions, ensuring that water availability does not become a limiting factor.
Consequently, in one-way coupling, the feedback from PCR-GLOBWB 2  to WOFOST is
inconsequential, as the continuous supply of water minimizes the need for dynamic interaction
between the models.
For rainfed conditions (Fig. 7), where water availability relies on green water only, crop yields
are comparatively lower than under irrigated conditions. Differences between the various
stand-alone, one-way, and two-way coupling approaches become apparent, particularly in the
western part of the CONUS. Notable differences in yields between stand-alone and two-way
coupling simulations are observed in maize and wheat crops, both under irrigated and rainfed
conditions. However, these differences are more pronounced for rainfed crops (see
supplementary Fig S12), where water availability is a crucial factor influencing crop yields. In
the case of rainfed soybeans, differences were less evident in all the models.
One-way coupling generally simulates higher yields for maize and wheat compared to two-
way coupling (see Fig 7). As described in Section 3.1 and Supplementary Information III, this
discrepancy arises from the transmission of soil moisture from the hydrological model to the
crop growth model in one-way coupling, without receiving feedback from crop development
to the hydrological model. This leads to an overestimation of late-season soil moisture
availability under drier conditions subsequently leading to a likely overestimation of simulated
crop yield by the one-way coupling. Clearly, this feedback is more important in the western
part of CONUS, which is likely related to larger interannual climate variability (with more dry
conditions) compared to the eastern part (see the section hereafter). The larger differences in
mean yields for rainfed crops, particularly in the western CONUS, that occur between one-way
and two-way coupled simulations are further illustrated by looking at the relative differences
between the two coupling methods (see Supplementary Information V; Fig. S10).

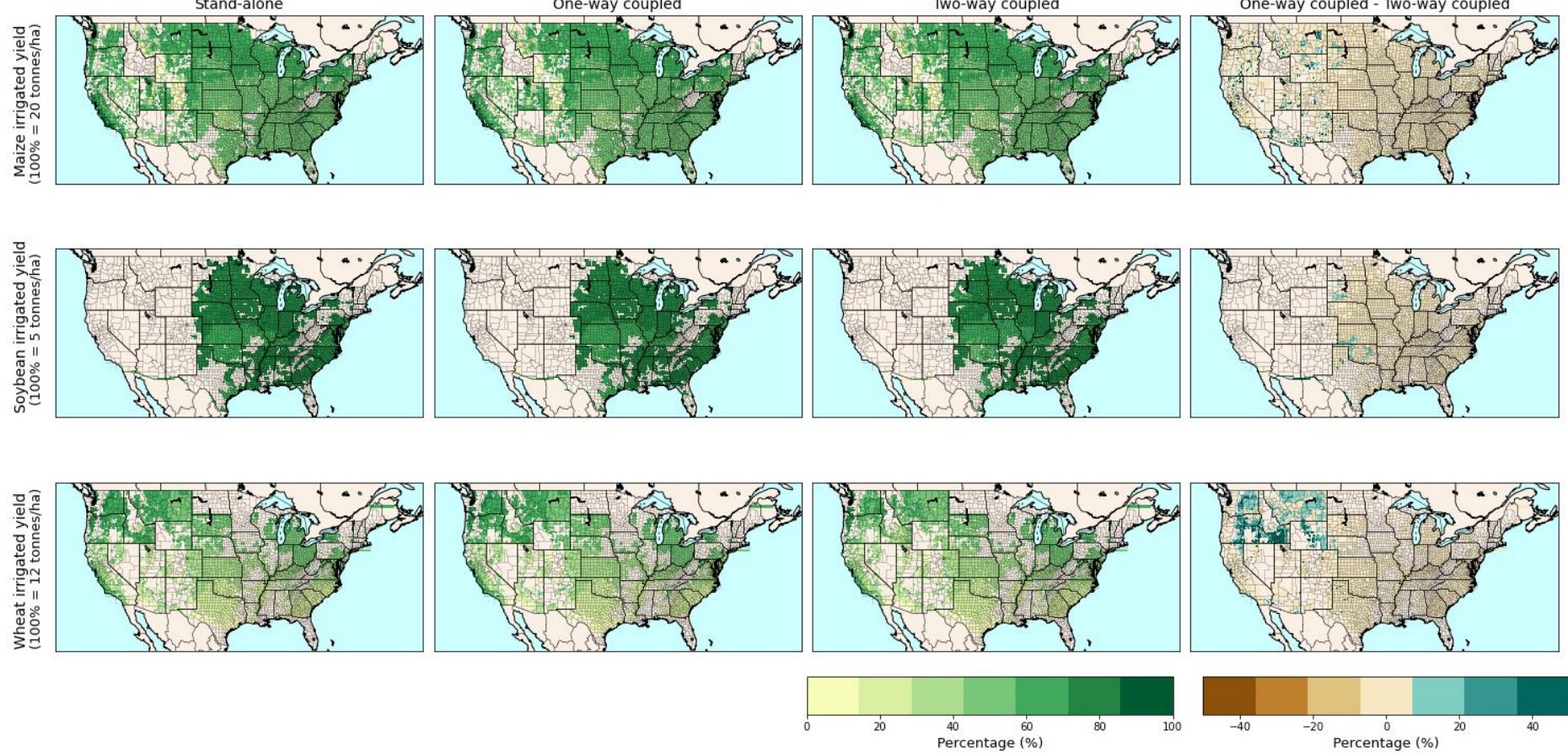

**Figure 6: Mean irrigated crop yields for maize, soybean, and wheat within CONUS as obtained from stand-alone, one-way and two-way coupled simulations and differences between one-way and two-way coupled simulations for 1979-2019. Legend in % of values shown on the y-axes.**




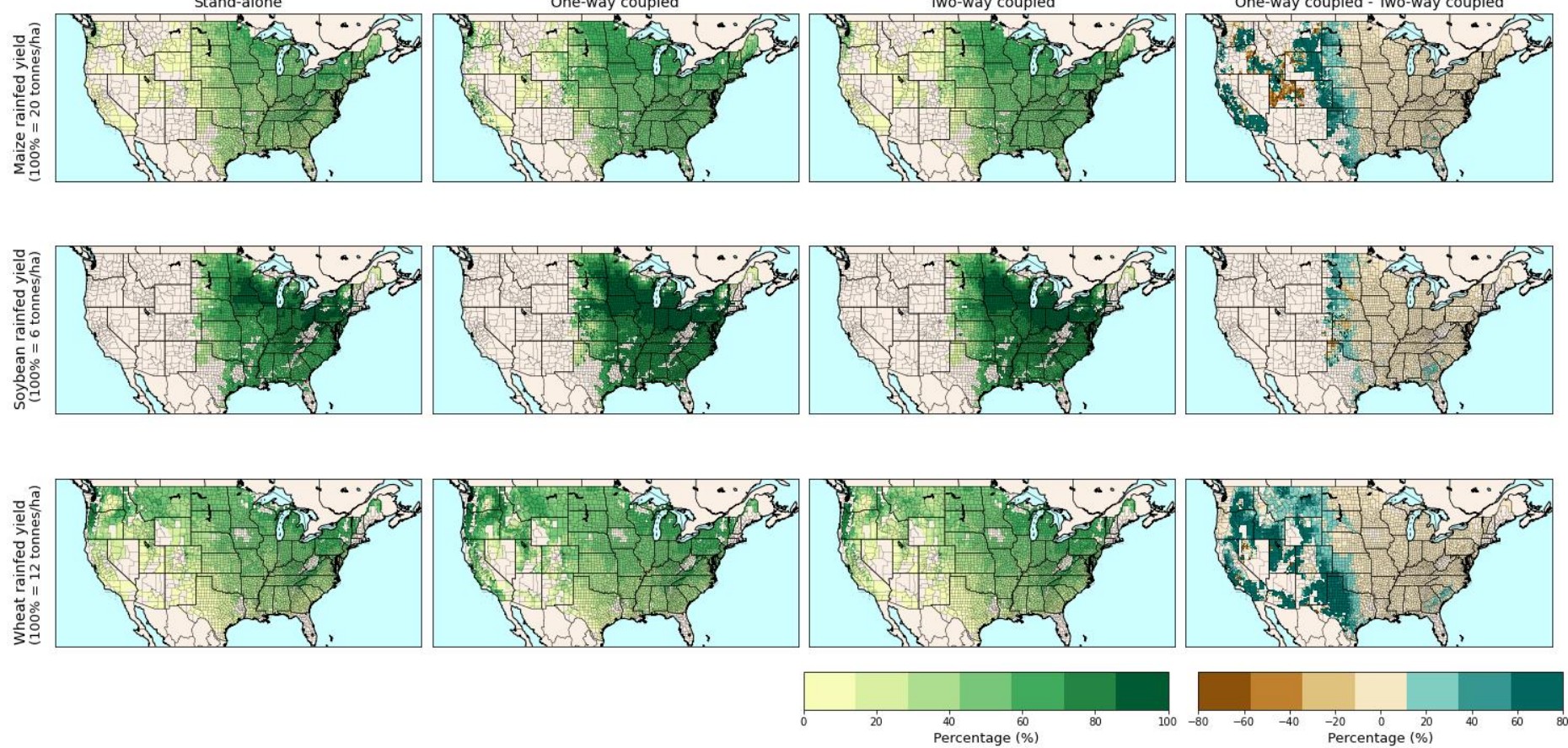


**Figure 7: Mean rainfed crop yields for maize, soybean, and wheat within CONUS as obtained from stand-alone, one-way and two-way coupled simulations and differences between one-way and two-way coupled simulation for 1979-2019. Legend in % of values shown on the y-axes.**



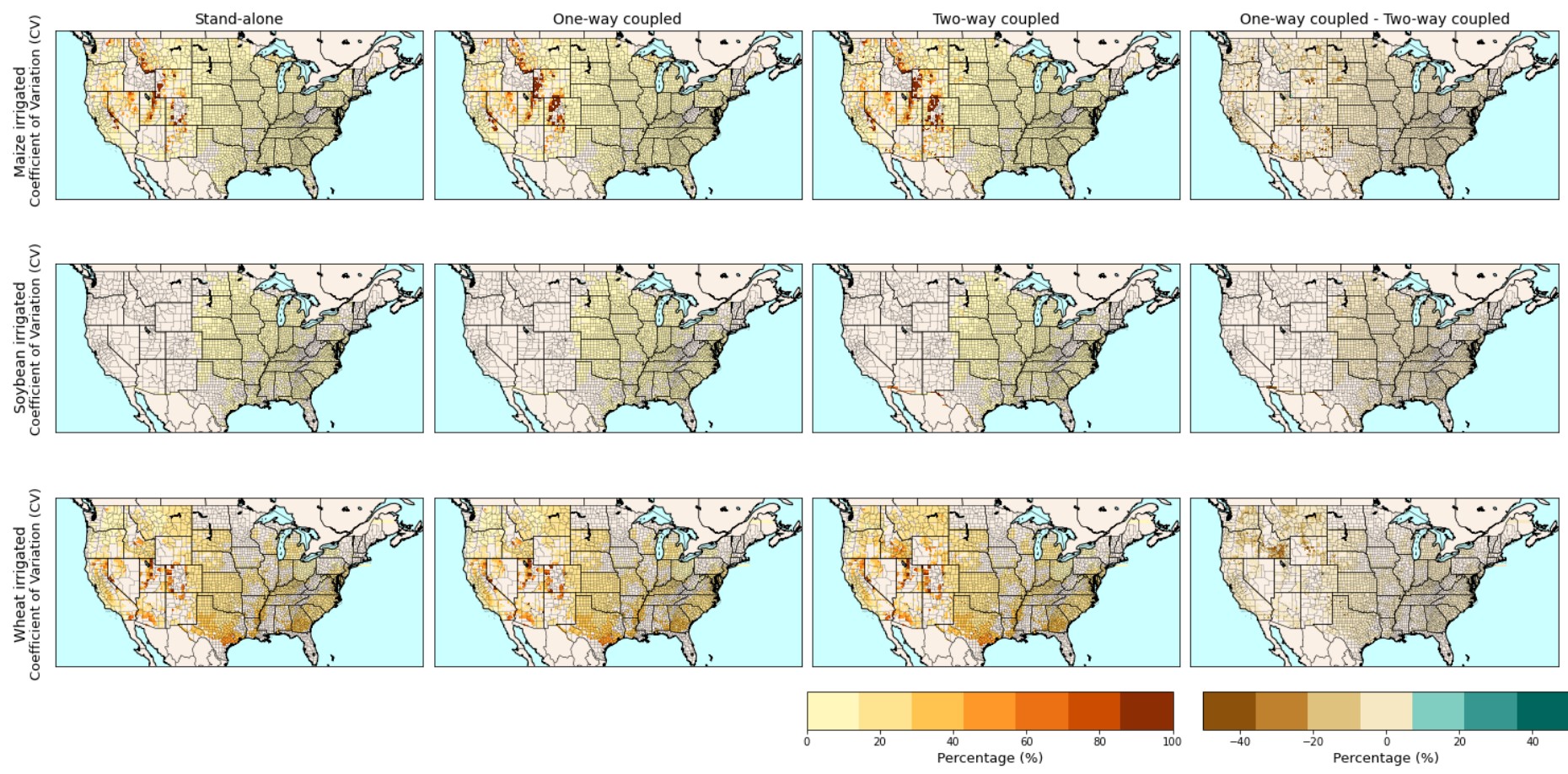



**Figure 8: Coefficient of Variation (CV) over 1979-2019 of irrigated crop yields for maize, soybean, and wheat within CONUS as obtained under stand-alone, one-**
**way and two-way coupling and difference between one-way and two-way coupling.**

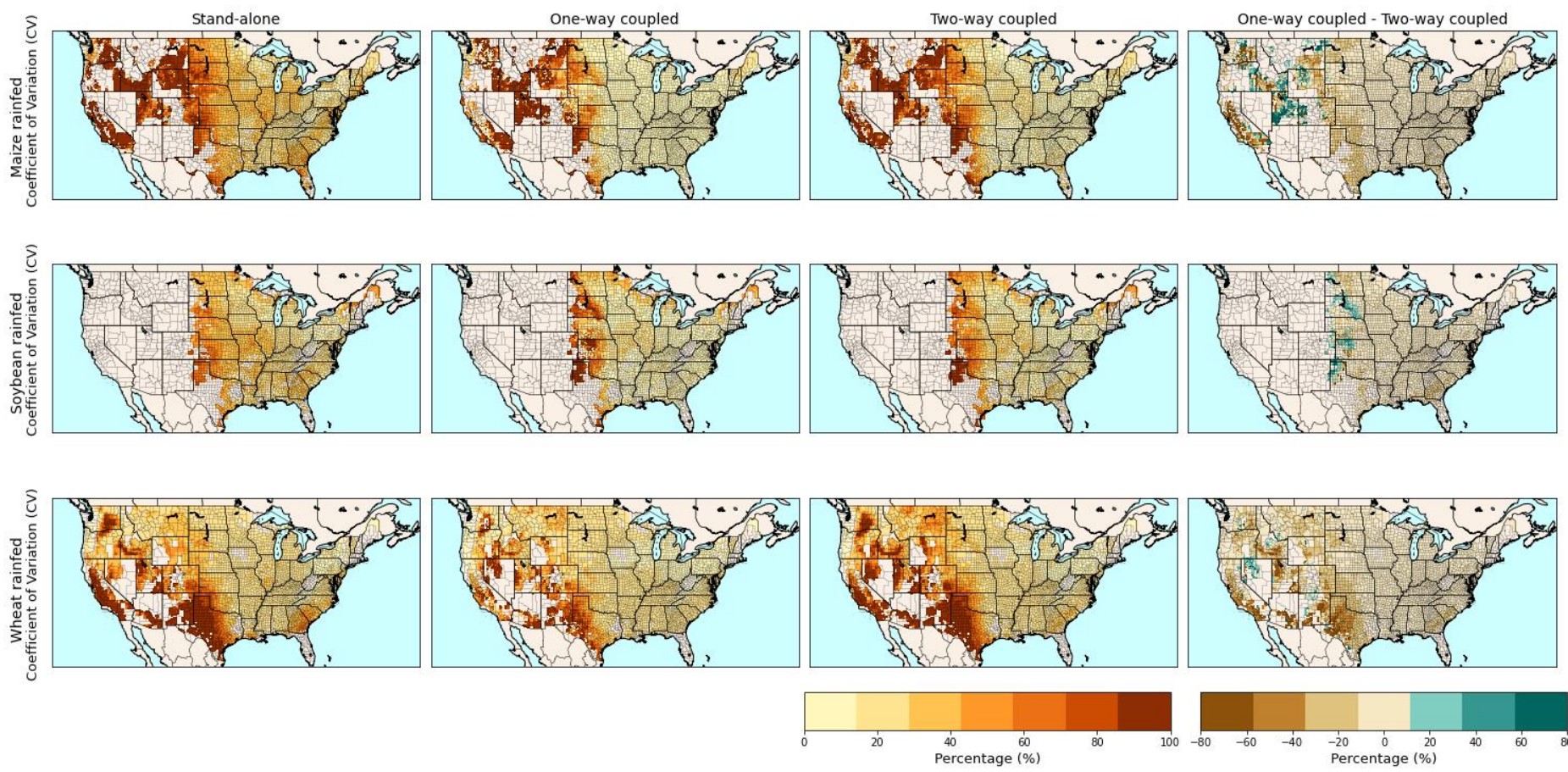


**Figure 9: Coefficient of Variation (CV) over 1979-2019 of rainfed crop yields for maize, soybean, and wheat within CONUS as obtained under stand-alone, one-way and two-way coupling and difference between one-way and two-way coupling.**

Spatial patterns of the coefficient of variation (CV) (in % of the mean) across CONUS for maize, soybean, and wheat are shown under irrigated (Fig. 8) and rainfed conditions (Fig. 9) comparing the simulations of the stand-alone, one-way, and two-way coupling. High CV values entail a larger inter-annual variability in crop yield.

In the eastern part of CONUS, the CV values both in irrigated and rainfed conditions are notably lower, suggesting a more stable and consistent pattern of crop growth in these regions. Conversely, in the mid-western and western CONUS, inter-annual variability is higher, owing to larger inter-annual climate variability in these parts. For irrigated crops, a larger CV is mostly apparent for maize and wheat. For a small number of instances, this could be caused by insufficient irrigation water availability during very dry and hot years, but most likely this is a temperature signal. Also, we note that in these parts of CONUS, some pixels have very low to minimal cropping areas, resulting in more pronounced fluctuations in yields. As can also be seen from Supplementary Information V Fig. S11, the differences between one-way and two-way coupled runs are generally small, except for some northwestern states.

Rainfed crops show larger values of CV, especially in the western part of CONUS, reflecting the larger sensitivity of rainfed agriculture to inter-annual climate variability (Fig. 9). It is also clear that the simulated inter-annual variability of simulated crop yield is larger for two-way than for one-way coupling, reflecting the importance of including crop phenology, in particular variation in rooting depth, when simulating available soil moisture. We also refer to Supplementary Information V Fig. S11 for relative differences between the two model coupling approaches. This larger inter-annual variability also partly explains the lower mean yields for rainfed crops and two-way coupling as was shown in Fig 7.

**3.4 Irrigation water use**

The scatter plot (Fig. 10) shows the relationship between reported USGS (after correction for area and irrigation efficiency – see 2.5.2) and simulated irrigation water withdrawals under one-way and two-way coupling. The plot shows that the simulated irrigation water withdrawals are correct in order of magnitude when compared to reported data across different states. The temporal variations (Fig. 11) illustrate that year-to-year changes in total irrigation water withdrawal over time are small for both one-way and two-way coupling and the reported totals.

Figures 10 and 11 show that irrigation water withdrawal is underestimated in total and for most states. The underestimation of irrigation water use by PCR-GLOBWB 2 was previously noted by Ruess et al., (2023). This underestimation was partly accounted for when using more

detailed crop cover data, irrigation efficacies and meteorological forcing than currently used in
the global version of PCR-GLOBWB 2.

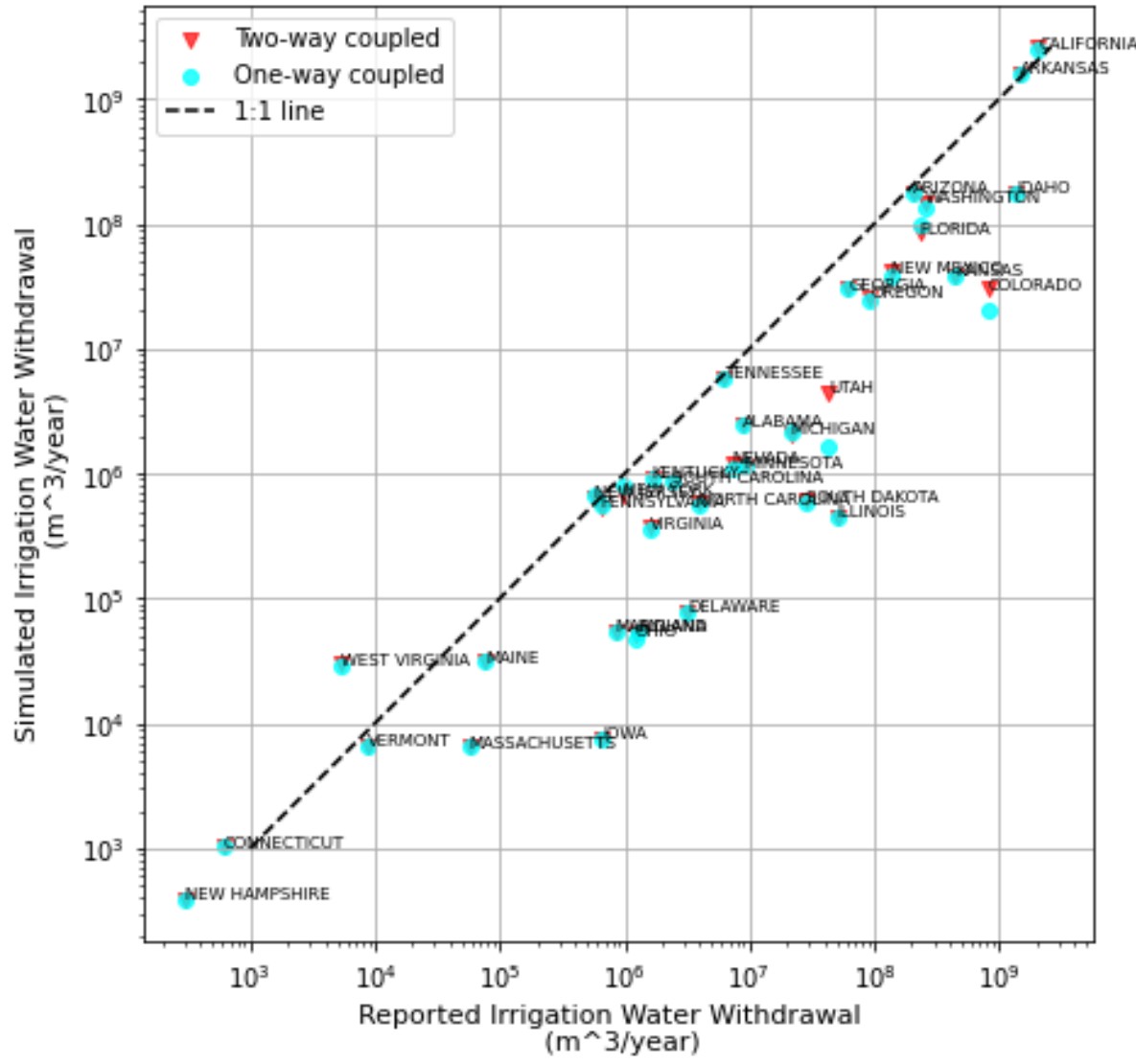


**Figure 10: Spatial variation of one-way and two-way irrigation water withdrawal compared with USGS-**
**reported water withdrawal data per state for all crops across the CONUS region with a logarithmic scale**

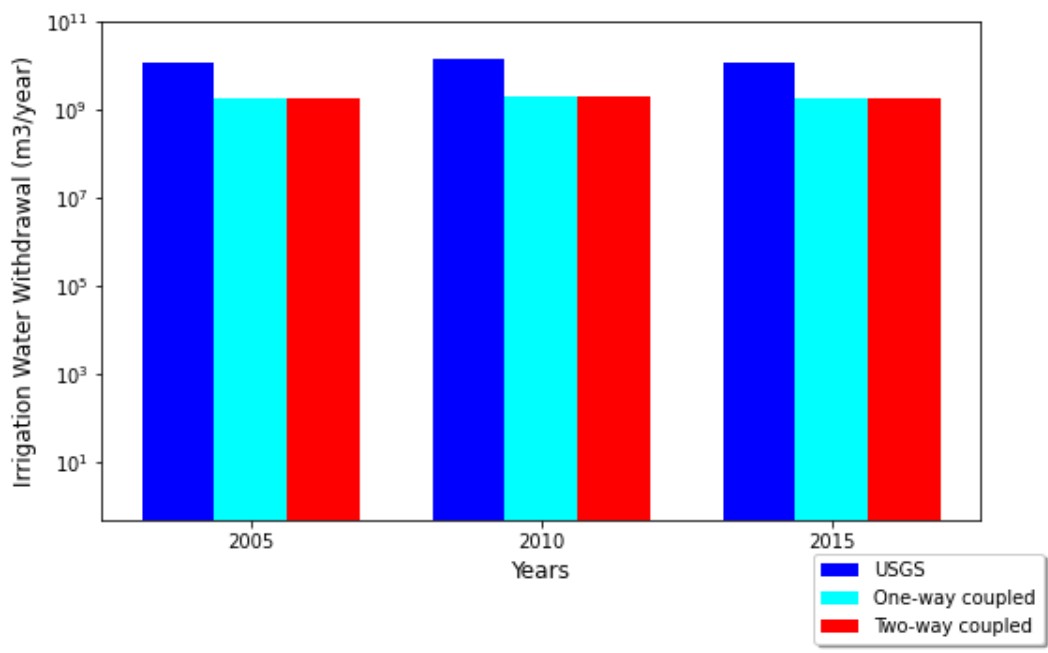


**Figure 11: Temporal variation of one-way and two-way irrigation water withdrawal compared with USGS water withdrawal data of 5-year intervals across the CONUS region with a logarithmic scale.**

## 4 Discussion and Conclusion

In this study, we developed a coupled hydrological-crop model framework to investigate the intricate feedbacks between water availability and crop growth within the CONUS region focusing on maize, soybean, and wheat. This discussion delves into the implications of the findings, emphasizing their significance and addressing both methodological considerations and inherent uncertainties.

We hypothesized that a more realistic representation of soil moisture dynamics and water availability will lead to better estimates of water stress and yield outcomes. Validation against reported yields however did not show a notable improvement compared to the stand-alone WOFOST, both for rainfed as well as for irrigated agriculture. Thus, if the focus is on yield only, coupling with a hydrological model such as PCR-GLOBWB 2 seems not needed. However, although not picked up by the validation exercise, the coupling still allows the inclusion of the impact of limited irrigation water availability as well as the impact of crop development on the hydrological system. Our study also shows that if the focus is on these impacts, it is necessary to use a two-way coupling to make sure that crop developments feeds back on evaporation and soil moisture.

832

Another hypothesis we tested is whether integrating real-time crop growth information into hydrological models will enhance the accuracy of predictions regarding irrigation needs and water resource allocation. Although it can be expected that feeding back crop information to PCR-GLOBWB 2 in the two-way coupling would improve estimates of irrigation water withdrawal, this could not be substantiated by comparison with reported water withdrawal statistics. One possible explanation is the use of constant crop area data across all years, which introduces uncertainties and limits the model's responsiveness to actual land-use dynamics.

The spatiotemporal analysis of hydrological impacts on crop growth confirms the results shown from the comparison with reported values. Notably, for rainfed crops, the estimated yield is mostly higher for one-way coupled simulations compared to two-way and stand-alone simulations. Also, the inter-annual variation of yield, that is, the sensitivity to drier and wetter years, is notably higher for the two-way coupled and stand-alone simulations than the one-way coupled simulations. This suggests that for a correct sensitivity to drought, a two-way coupling that includes the feedback of crop status to the hydrological system is needed.

Our studies adds to previous work by Droppers et al., (2021), which investigated worldwide water constraints and sustainable irrigation by coupling the Variable Infiltration Capacity (VIC) hydrological model with WOFOST and Zhang et al. (2021) who focused on refining the coupled VIC hydrological model with a crop growth model EPIC by incorporating the evapotranspiration module at a regional scale. In comparison, our research extends the analysis to a finer spatial scale and places a stronger emphasis on the comprehensive integration of feedback loops between hydrology and crop growth. Particularly, we demonstrate the importance of two-way coupling in capturing realistic yield outcomes, which is particularly evident for rainfed crops. This is mainly because the two-way coupled system addresses the influence of crop status on evapotranspiration and rooting depth, thereby impacting soil moisture content, which in turn feed backs on crop growth. The two-way coupling approach provides a more realistic depiction of water availability for crops, which results in larger inter-annual variability and lower mean crop yields when inter-annual climate variability is significant. Including this two-way interaction is particularly important under drier conditions (see section 3.2) or if the coupled framework is used to assess reduced surface water availability under climate change or the impact of environmental constraints on groundwater and surface water use.

While the results of this study offer valuable insights into the coupled hydrological-crop model framework, it is essential to recognize and address the uncertainties associated with the structure and parametrization, as well as inherent limitations in the research. A significant limitation is that the study does not account for potential advancements in agricultural technology and evolving farming practices, which could impact crop yields, This becomes evident when comparing yield estimates with observations over time (section 3.1; Fig. 4).

Furthermore, uncertainties linked to input datasets (Porwollik et al., 2017; Roux et al., 2014) such as crop calendars, cultivars and land-use changes introduce potential limitations and implications for the study results. Accurate representations of crop growth dynamics hinge on accurate crop calendar definitions (Wang et al., 2022), encompassing planting, maturation, and harvesting periods. Variations in these timelines due to climate change or evolving agricultural practices potentially introduce uncertainties in yield predictions. Additionally, the assumption of static cultivars neglects potential shifts in agricultural practices or the introduction of new varieties, influencing crop growth responses to environmental stressors over time. Land-use changes further contribute to uncertainties (Prestele et al., 2016; Eckhardt et al., 2003; Dendoncker et al., 2008) as dynamic shifts in agricultural practices alter water demand, evapotranspiration patterns, and overall hydrological dynamics. Ignoring these potential shifts limits the model's ability to capture the complex interactions between water and crop systems, and this should be considered in future development steps.

Hence, future work should also consider representing the dynamic nature of crop areas, including both irrigated and rainfed crop harvest areas, as well as the total crop area. The assumption of constant areas, as made in prior studies (Müller et al., 2017; Ai and Hanasaki, 2023; Jägermeyr et al., 2021) was based on data availability constraints, but acknowledging the potential variability in these factors over time. Addressing this aspect is crucial for enhancing the accuracy of yield calculations and, consequently, advancing the overall understanding of hydrological-crop growth interactions. The integration of such variability into modelling frameworks is not only essential for improving the accuracy of assessments but also for contributing to an enhanced understanding of the broader water-food nexus.

In conclusion, the development and application of the two-way coupled hydrological-crop growth model framework presented in this study represents a significant advancement in our ability to understand the cascading mechanisms and feedbacks between water and crop systems. Although it does not show an improvement of yield estimates per se, the coupling

framework enhances our understanding of the interplay between hydrology and crop growth. Also, through the sectoral water use modules of PCR-GLOBWB 2, it contains the necessary components to evaluate large-scale water use management strategies, and simulate the large-scale impacts of informed decision-making under change, particularly when dealing with hydroclimatic extremes.

**Author contribution**

SC designed the study, performed the analyses, validation, and visualization of the results under the supervision of LPHvB, MTHvV, and MFPB. SC developed the coupled framework in close collaboration with LPHvB. JA contributed to the conceptualization of software. SC wrote the original draft manuscript and all co-authors reviewed and edited the manuscript.

**Code and data availability**

The developed coupled PCR-GLOBWB 2-WOFOST model framework is available at https://zenodo.org/doi/10.5281/zenodo.10681452. The datasets used in the coupled model framework are available at https://opendap.4tu.nl/thredds/catalog/data2/pcrglobwb/version_2019_11_beta/pcrglobwb2_input/catalog.html.

**Competing interests**

The contact author has declared that none of the authors has any competing interests.

**Acknowledgement**

The authors acknowledge dr. Bram Droppers (Utrecht University) and dr. Iwan Supit (Wageningen University) for their valuable advices on the WOFOST crop model.

**Financial support**

This research has been funded by the European Union Horizon Programme GoNexus project (Grant Agreement Number 101003722). MTHvV was financially supported by the Netherlands Scientific Organisation (NWO) by a VIDI grant (VI.Vidi.193.019) and the European Research Council (ERC) under the European Union's Horizon Europe research and innovation program (grant agreement 101039426 B-WEX).

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
