# Peer review of "Relevance of feedbacks between water availability and crop"

_EGUsphere, 2024_

## Author Response (AR1)

**Response to reviewer's comments**

Dear Editor,

Thank you very much for handling our manuscript. We sincerely thank all the referees for their time, valuable comments, and suggestions, which have helped us to improve our manuscript. We have carefully revised the manuscript to include the inputs from the referees and are attaching the updated version. We hope that these revisions make our work acceptable for publication in HESS. Please find below a point-to-point response to each of reviewers comments.

**Response to Reviewer 1 comments**

We thank Referee 1 for their thorough review, which has helped to improve our paper.

**Summary: This study couples a global hydrological model and a crop growth model in both one-way mode (hydrological model provides soil water storage to crop model) and two-way mode (additionally, crop model provides land surface fluxes, LAI, and rooting depth to hydrological model). This is a noteworthy technical achievement, and the developed code is openly shared. The chosen coupling approach is not discussed and justified much, and possible alternatives are not investigated and compared. Reported findings are largely dependent on modeling technicalities related to the chosen coupling approach. Moreover, it is unclear which specific phenomena / research questions require the use of a coupled hydrological / crop modeling approach. My recommendation would be to publish this as a technical note, either in HESS or a journal specialized on advances in environmental modeling and software after revisions.**

Thank you for the comments. We have added a new section in the Methodology that justifies our chosen coupling approach and expanded the Introduction to clearly describe why this specific coupling is required to address our research objectives. Below, we provide the details of the updated sections.

While we appreciate the recommendation to publish as a technical note, we aim to keep this paper focused on the scientific objectives while also incorporating the necessary technical details. We have revised the manuscript accordingly to ensure it covers both the scientific and technical aspects.

We have added a section titled "2.2. Justification of model coupling" to the Methodology (lines 305-417 in the manuscript with track changes). We have also moved the "Implementation of the framework coupling" text from the supplementary material to this section (2.2). Expanded the Introduction with additional paragraphs (lines 136-150 in the manuscript with track changes) and improved the clarity of the research questions (lines 127-133 in the manuscript with track changes).

**Specific comments:**

1. **How is reservoir release and inter-basin transfer handled in PCR-GLOBWB? From the text, it seems as if irrigation water availability is an exogenous boundary condition provided by PCR-GLOBWB. However, in reality, there is probably a feedback mechanism between crop state and reservoir management, i.e. system managers will be responsive to crop state and will adjust reservoir release and transfer decisions.**

We are aware that inter-basin transfers can have regional impacts but they are poorly defined at the global scale and difficult to account for in the daily dynamics of PCR-GLOBWB 2. As such, inter-basin transfers are disregarded for this assessment.

Reservoir operations in PCR-GLOBWB are either defined in terms of hydropower generation or the water supply. These operation types are mutually exclusive. In our simulation with PCR-GLOBWB 2, we considered all reservoirs to be of the hydropower type, meaning that reservoir storage is maximized and the average discharge passed on as much as possible. The alternative, using the reservoir for water supply, would be more adequate; in that case, the amount of water that is released is intended to match the downstream demand of all sectors. In future simulations, we will adopt this alternative reservoir operation scheme and include the feedback between crop state and reservoir storage, as mentioned by the reviewer. However, in this case, we opted not to include this yet as it requires information on the irrigation command areas (irrigation districts). This requires additional input and parameterization and adds uncertainty that we want to evaluate rigorously in terms of the provenance of irrigation water (e.g., surface vs groundwater). In our current setup, the irrigation water demand can be met fully by either surface water and groundwater and we compared this lumped with the total irrigation water requirements as reported by the USGS which, because of the fact that these are five-year estimates, are insufficiently detailed to highlight short-term variations. To pick this up, validation should look in detail at the aforementioned irrigation districts, but such a deep dive into a more local and more event-based nature is beyond the scope of our present manuscript.

2. **Much discussion focuses on the importance of two-way coupling for the cropped areas, i.e. providing updated rooting depth, LAI and ET to PCR-GLOBWB, which are responsive to water stress, particularly on the rainfed areas. This leads to the question if such feedback mechanisms should not also be modelled on the non-agricultural portions of the landscape in PCR-GLOBWB?**

The WOFOST crop growth model used in this study is specifically designed to simulate annual field crops, including maize, soybean, and wheat. In our framework, WOFOST calculates the vegetation states, (such as leaf area index (LAI), biomass and root depth) and fluxes (e.g. evapotranspiration) exclusively for these irrigated and rainfed crops. For all other land cover types, including non-agricultural areas, the states and fluxes are simulated within PCR-GLOBWB 2. This approach allows us to focus on the two-way

coupling feedback mechanisms on the specific crops of interest, while other vegetation and non-vegetation fluxes are appropriately handled within PCR-GLOBWB 2.

Moreover, the coupled framework is designed with the flexibility to integrate a vegetation or non-vegetation simulation model in the future, ensuring that the current structure remains intact. This means that while the current study focuses on specific crops, the framework can be expanded to include more feedback mechanisms for other land-cover types as a potential next research step.

We have added the explanation of vegetation and non-vegetation simulation in the Methodology (lines 453-461 in the manuscript with track changes).

3. **The chosen coupling approach through the variables of soil water storage and land-surface fluxes is postulated without much discussion of possible alternatives. Why not couple the two models through irrigation and percolation rates only, i.e. let WOFOST handle the soil moisture balance? Is this because phreatic ET could be relevant in some cropped system, i.e. you want to capture direct groundwater use of crops?**

The decision to couple the models through soil water storage and land-surface fluxes, rather than just through irrigation and percolation rates, was carefully considered. WOFOST uses a classic water balance approach suited for freely draining soils, where groundwater is too deep to affect soil moisture content in the rooting zone. This approach divides the soil profile into two compartments: the rooted zone and the lower zone extending from the actual rooting depth to the maximum rooting depth. The subsoil below this maximum rooting depth is not considered. As roots extend deeper towards the maximum rooting depth, the lower zone gradually merges with the rooted zone. This approach is suitable for regional applications with limited soil property information. Soil moisture in the root zone serves as a primary link between the WOFOST model and the underlying soil module.

However, while this approach is effective for basic applications, it has limitations in capturing the full dynamics of soil moisture availability and its interaction with the broader hydrological cycle. It does not account for more complex processes such as lateral flows or deeper percolation, which can significantly impact crop water availability.

By using PCR-GLOBWB 2 to handle the soil moisture balance and land-surface fluxes, we ensure a more accurate representation of soil water dynamics, including important processes like phreatic evapotranspiration (ET), which can be relevant in certain cropping systems where groundwater use by crops is significant. This coupling approach allows us to account for complex interactions between crop growth and the hydrological system, which would be difficult to achieve with WOFOST stand-alone.

We have added details about the WOFOST structure to the WOFOST description section in the Methodology (lines 266-304 in the manuscript with track changes). The decision to couple the models through soil water storage and land-surface fluxes, rather than just

through irrigation and percolation rates is explained in the justification section in Methodology (section 2.2), please see the response of the summary comment.

**4. Related to the previous comment: It would be valuable to explain on a more intuitive basis, what kind of feedback mechanisms should be investigated with this modelling system and why the chosen coupling approach is the most appropriate for these purposes. As far as I can see, crop yield responses to water and heat stress can be simulated in stand-alone mode with irrigation allocations provided by PCR-GLOBWB. Is it the influence of groundwater/phreatic ET that is in focus here?**

The expected feedback mechanism between the hydrological and crop models is outlined in the Introduction section. Please refer to our response of your summary comment.

We hypothesize that the feedbacks between hydrology and crop growth are significant and complex. Changes in soil moisture and water availability are expected to directly influence crop water uptake, growth rates, and yield outcomes. Conversely, crop processes such as evapotranspiration and root water uptake are likely to impact soil moisture levels, groundwater recharge, and surface water flows, thereby altering the water resources. Furthermore, we anticipate that the integration of real-time crop data into hydrological models will enhance the accuracy of predictions regarding water stress, irrigation needs, and crop productivity.

As discussed in the previous comment 3, while crop yield responses to water and heat stress can be simulated in a stand-alone model with irrigation allocation, this approach does not fully capture the intricate interactions between crop growth and hydrological systems. Coupling PCR-GLOBWB 2 and WOFOST through soil water storage and land-surface fluxes addresses this limitation by incorporating the feedback loops between the crop processes and hydrological factors, such as surface and irrigation water availability.

**5. Figs 4/5, 6/7 and 9 – please add results from stand-alone simulations also for reference / comparison.**

For better clarification, we have included the results of the stand-alone model as well as the differences between one-way and two-way coupled approaches in Figures 4, 5, 6 and 7, which are referred to as Figures 6, 7, 8 and 9 in the revised manuscript with track changes (lines 797-818)

However, in Figure 9, which illustrates the temporal variations in one-way and two-way irrigation water withdrawals, we did not include the stand-alone results. This is because the stand-alone WOFOST model simulates potential production based on the assumption that soil moisture content is at field capacity, and it only calculates the actual evaporation and transpiration rates under these conditions. Therefore, the stand-alone results are not directly comparable to the coupled approaches in terms of irrigation water withdrawals.

**6. Fig 9: I wonder why a logarithmic y-axis is used. It seems that differences between USGS data and models are a factor 10 or so… is that a good result?**

Thank you for your observation regarding the use of a logarithmic y-axis in Figure 9, which is Figure 11 in the manuscript with the track changes. The logarithmic scale was chosen to effectively capture and visualize the wide range of irrigation water withdrawals reported by USGS and simulated by our model framework, especially across different states and crop types. While the differences between USGS data and our model outputs can indeed be as large as a factor of 10, this is primarily due to the necessary corrections for area and irrigation efficiency.

As explained in the methodology section (2.4.2), now referred to as section 2.5.2 in the revised manuscript), the USGS data represents total irrigation water use for all crops, while our model is based on four crops from MIRCA2000. To ensure a meaningful comparison, we corrected the simulated withdrawals to account for differences in total irrigated area and average irrigation efficiency of all crops with the four crops considered in this study. However, discrepancies still arise because we applied a uniform efficiency correction across diverse crop types, which may not fully reflect the lower efficiencies typically associated with annual field crops compared to high-efficiency cash crops.

Thus, while the model performs well in capturing overall trends, some differences are expected, especially when aggregated across large and diverse regions such as CONUS. The logarithmic scale helps in highlighting these variations, providing a clearer view of how the model and observed data compare across a broad spectrum of values.

7. **Figure 8: Please add 1:1 line for orientation.**

Thank you for the suggestion. We have revised Figure 8 accordingly (now Figure 10 in the manuscript with track changes, lines 861-863 ).

8. **Modelling uncertainties: It would be very instructive if confidence bands could be added to some of the simulation results (e.g. Figures 2,3). Differences between simulations, especially between one-way coupling and two-way coupling, are often quite small and probably insignificant compared to overall modeling uncertainty.**

Thank you for the suggestion. We acknowledge the importance of addressing modelling uncertainties. While the differences between one-way and two-way coupling may appear small at a large scale, they become more pronounced at local levels. The averaging process tends to diminish these differences when analyzed over broader regions, which is why they might seem less significant.

While confidence bands can offer insight into uncertainty ranges, they might not accurately represent the localized variability and uncertainties that differ from one region to another. Instead of broad confidence bands, we focused on the differences in coupling approaches and their implications on the model's output. Future studies could explore localized uncertainty analyses to provide insight on modelling uncertainties

**Response to Reviewer 2 comments**

We thank Referee 2 for their thorough review, which has helped to improve our paper.

**Summary:**
**This study presents a coupled hydrological and crop model that integrates PCR-GLOBWB 2 with WOFOST, effectively accounting for the interactions between hydrology and crop growth. The work focuses on how two-way interactions and feedback mechanisms between crop growth and hydrological systems would benefit the modeling. The authors show that this coupled framework can reproduce crop yields well, and highlight an improvement in performance when there is two-way coupling compared to one-way coupling. Overall, the study is clear and the model is open source. However, my main concern is that the paper focuses on presenting results from a specific coupling scheme without adequately discussing the broader implications or conducting a comprehensive sensitivity analysis of alternative schemes. While the paper is detailed as a technical report describing the operational aspects of the model, it falls short in exploring how different coupling schemes might affect our understanding of the feedbacks - a scientific point that the title suggests. Such a more thorough analysis would provide invaluable insights and greatly benefit future research on similar models.**

**Given the current scope of the paper, I recommend revising the title, abstract, and introduction to better reflect its focus on technical development (or specific reporting on a particular coupled model) rather than broader scientific (and methodological) discussions. Below are some specific suggestions for further improvement.**

Thank you for the comments and the suggestions to publish the paper as a technical note. While we appreciate the recommendation, we prefer to keep the paper focused on the scientific objectives while also incorporating the necessary technical details. We have revised the manuscript to ensure it covers both the scientific and technical aspects.

The revision includes as follows:

1. Introduction section: We have expanded the Introduction section to better address the scientific objectives and incorporate feedback to align with the paper's title (lines 127-133; 136-150 in the manuscript with track changes)
2. Methodology section: We have made a major revision to the Methodology section, particularly in justifying the chosen coupling approach (section 2.2 titled Justification of model coupling, lines 305- 417 in the manuscript with track changes). Additionally, we have moved the technical details of the coupling framework from the supplementary material to the Methodology section 2.2. However, we did not include a sensitivity analysis of the models. This decision was based on the fact that both the models, hydrological and crop growth, used in the study have been extensively applied and tested across local to global scales. The

model parameters are already fine-tuned, as documented in the description of the model section in Methodology (section 2.1 in the manuscript with track changes).

3. Results sections: We have replaced Figures 4, 5, 6, and 7 in the Results section. We now compare the stand-alone, one-way and two-way coupling approaches, along with the differences between the one-way and two-way approaches, for better visualization. The text in the results section has also been revised to better address the differences between the approaches and the relevant feedback between the WOFOST crop growth model and PCR-GLOBWB 2 hydrological model. The revised figures are now referred to as Figures 6, 7, 8 and 9 in the manuscript with track changes (lines 798-819; revised text in lines 595-693; 754-795).

We believe these revisions strengthen the manuscript by providing a detailed description of both scientific and technical aspects of the work. Please refer to the revised manuscript with track changes for detailed updates.

**Specific comments:**

**1) WOFOST is a crop simulation data, I wonder how non-crop vegetation is considered in the model in the two-way coupling where phenologies are calculated by a crop simulation model. Please clarify.**

The WOFOST crop growth model used in this study is specifically designed to simulate annual field crops, including maize, soybean, and wheat. In our framework, WOFOST calculates the vegetation states, (such as leaf area index (LAI), biomass and root depth) and fluxes (e.g. evapotranspiration) exclusively for these irrigated and rainfed crops. For all other land cover types, including non-agricultural areas, the states and fluxes are simulated within PCR-GLOBWB 2. To be more specific, for the fraction of land cover that is different from maize, wheat and soybean, the vegetation states and fluxes are calculated within the PCR-GLOBWB 2. For these land cover types, vegetation phenology in the form of crop factors, is approximated by a yearly climatology. This approach allows us to focus the two-way coupling feedback mechanisms on the specific crops of interest, while other vegetation and non-vegetation fluxes are handled within the broader PCR-GLOBWB 2 framework.

We have added the explanation of vegetation and non-vegetation simulation in the Methodology (lines 453-461 in the manuscript with track changes).

**2) L197: The meaning of "astro" here is not clear**

"Astro" refers to the Astronomical module, which calculates the day length, several intermediate variables for determining solar elevation, the integral of solar elevation over a day, and the fraction of diffuse radiation. The Astronomical module is updated in the WOFOST description section (line 273 in the manuscript with track changes) and also shown in revised Figure 1 (line 194).

**3) L390-406: It would be helpful to clarify why stand-alone WOFOST can produce a similar simulation with two-way coupling.**

The similarity in results between the stand-alone model and the two-way coupling has been clarified in the revised Results section (lines 625-652 in the manuscript with track changes).

**4) Section 3.2: If possible, the content can be merged with Section 3.1, and some metrics in Table 1 can be directly shown in Figure 2.**

Thank you for the suggestion to merge Section 3.2 with Section 3.1 and to incorporate some of the metrics from Table 1 directly into Figure 2. After careful consideration, we believe that keeping Section 3.2 as a separate section is important for maintaining the clarity and structure of our analysis. Section 3.1 focuses on comparative analysis, while Section 3.2 provides insights into evaluation statistics. Merging these sections could dilute their respective focus and make the presentation of our findings less coherent.

Regarding the suggestion to incorporate metrics from Table 1 into Figure 2, we decided to keep these metrics in a table format because they provide detailed quantitative insights that are best presented in tabular form. Including them in Figure 2 might overcrowd the figure and take away from its visual clarity.

We believe that the current structure best serves the purpose of clearly and effectively communicating our findings.

**5) Figures 4 and 5, it's hard to see the difference between one-way and two-way from the color scales, and I don't think the difference is "notably" (line 471). I would suggest an additional visualization to better compare the simulated crop yields.**

Thank you for the suggestion. For better visualization, we have revised figures 4 and 5 (figure 6 and figure 7 in the revised manuscript with track changes; lines 798-807) and included panels on the differences between one-way and two-way coupled approaches.

**Response to Reviewer 3 comments**

We thank Referee 3 for their thorough review, which has helped to improve our paper.

**This study analyzed the development of a coupled hydrology-crop model framework to investigate the intricate feedbacks between water availability and crop growth within the CONUS region focusing on maize, soybean, and wheat. The PCR-GLOBWB hydrological model was coupled with the WOFOST crop growth model to quantify both the one-way and two-way interactions. The result in the temporal and spatial variation of crop yield whether including interactions is interesting.**

**However, it is difficult to explain how the effect of one-way or two-way interaction is different as a result of discussing feedback, and for this reason, this is explained by the technical note. I recommend major revision after revising the title, abstract, and result to better introduce technical development of this study. This paper is a useful description in the hydrology and crop growth modeling technical area.**

Thank you for your feedback and recommendations. We understand your concerns about the clarity of the differences between one-way and two-way interactions. In our revised manuscript, we have provided a detailed explanation of these effects, as outlined in our responses to your comments below. We have carefully discussed the feedback mechanisms and their implications within the scientific framework, ensuring that the technical aspects are well-integrated with the scientific analysis.

We believe that the current structure of the paper, which balances both the technical development and the scientific exploration of the feedback mechanisms, is crucial for advancing understanding in the field of hydrology and crop growth modeling. While we appreciate the suggestion to focus more on technical development, our intention is to maintain this manuscript that addresses both the scientific and technical dimensions of the study.

**Detailed comments are below.**

1. **Overall, it is difficult to clearly identify the difference between one-way and two-way in result section. Can you discuss about the result from the difference between one-way and two-way method? It is difficult to discuss the feedback-based scientific point as to what feedback is due to the two-way interaction.**

The main difference between the one-way and two-way coupling methods lies in the feedback mechanisms that are captured. In the one-way coupling, data exchange is only from the hydrological model (PCR-GLOBWB 2) to the crop model (WOFOST), without any feedback from the crop model back to the hydrological model. This means that while crop growth and yield are influenced by water availability as simulated by the hydrological model, the crop model does not influence soil moisture, groundwater levels, or surface water flows.

In contrast, the two-way coupling allows for feedback from the crop model to the hydrological model. This feedback is critical in capturing the dynamic interactions between crop growth and water resources. For example, crop water uptake, evapotranspiration, and root growth as simulated by WOFOST directly influence soil moisture levels, groundwater recharge, and surface water availability in PCR-GLOBWB 2. These interactions are essential for accurately representing the impacts of crop growth on water resources and vice versa.

In irrigated conditions, similar yields were observed between the one-way and two-way coupled approaches. This is expected since soil moisture is kept at optimum levels under irrigated conditions, ensuring that water availability does not become a limiting factor. Consequently, in one-way coupling, the feedback from WOFOST to PCR-GLOBWB 2 is inconsequential, as the continuous supply of water minimizes the need for dynamic interaction between the models.

In rainfed conditions, one-way coupling tends to simulate higher yields compared to two-way coupling. This discrepancy arises from the transmission of soil moisture from the hydrological to the crop growth model in one-way coupling, without receiving feedback from crop development to the hydrological model. As stated before, this may overestimate soil moisture availability under drier conditions subsequently leading to a likely overestimation of simulated crop yield by the one-way coupling. Clearly, this feedback is more important in the western part of CONUS, which is likely related to larger interannual climate variability (with more dry conditions) compared to the eastern part.

Additionally, the two-way coupling reveals that during dry spells, the interaction between declining soil moisture and crop growth leads to an earlier onset of crop stress. In contrast, one-way coupling which does not account for feedback from crop stress to soil moisture, tends to overestimate the severity and timing of water stress on crops. In the two-way coupling, the slower crop development due to water stress in dry years feeds back into the hydrological cycle by reducing evapotranspiration rates. This reduction in evapotranspiration helps to conserve soil moisture, thereby influencing the hydrological model's predictions of soil moisture availability. Such feedbacks are absent in one-way coupling, where the fixed phenology leads to an overestimation of water uptake by crops, further exaggerating yield estimates. In some regions of the western CONUS, one-way coupling underestimates yields for rainfed crops of maize and wheat compared to two-way coupling, as the crop growth model WOFOST does not influence hydrological processes in the one-way coupling.

We have updated the objectives in the Introduction section (lines 127-133, in the manuscript with track changes). Figures 4, 5, 6 and 7, which illustrate one-way and two-way coupling have been revised to include an additional panel showing differences between one-way and two-way coupling approaches. The corresponding text in the results section has also been updated. The revised figures are now referred to as Figures 6, 7, 8 and 9 (lines 798-819) and the updated text can be found in lines 744-795 of the manuscript with track changes.

2. **How about showing additional difference figures between one-way coupled and two way coupled (Fig. 4- 7)?**

We have included the results from the stand-alone model as well as the differences between one-way and two-way coupled approaches in Figures 4, 5, 6 and 7 (referred to as Figures 6, 7, 8 and 9 in the manuscript with track changes)

3. **How about express Table 1 as a bar graphs like Fig.3? It would be easier to compare the differences if you provide the numbers with the figure**

Thank you for the suggestion to express Table 1 as bar graphs similar to Figure 3. We agree that visual representations can sometimes make comparisons easier. However, we believe that the numerical data in Table 1 are best presented in a tabular format. This allows for a more precise and detailed comparison of the values, which might be less clear when presented as bar graphs.

4. **For the convenience of the reader, it would be nice if you could refer to the figure subsection in the result section**

We agree that referring to the figure subsections in the results section would enhance readability. We have revised the text where necessary. We believe the remaining part of the text is properly referred in the previous version. The revised text can be found in the results section, lines 673 and 694 in the manuscript with track changes.

5. **How about adding diagonal line to Fig. 8?**

Thank you for this suggestion. We have added a one-to-one diagonal line to Figure 8 (referred to as Figure 10, lines 861-863 in the manuscript with track changes).

6. **In Figure 2, whether it is possible to compare the performance through interannual correlation with the reported field?**

Thank you for bringing this up. However, the correlation analysis is already thoroughly presented in Section 3.2 of the manuscript (Table 1). Incorporating the correlation analysis directly into Figure 2 would likely lead to overcrowding and might reduce the clarity and effectiveness of the figure. We believe that keeping the correlation analysis presented with the other performance metrics (normalized RMSE and normalized BIAS) in Table 1 separated from Figure 2 maintains the focus and clarity of both the figure and the analysis.

**Response to Reviewer 4 comments**

We thank Referee 4 for their thorough review, which has helped to improve our paper.

**The work by Chevuru et al. developed a couple hydrology (PCR-GLOBWB 2) – crop growth (WOFOST) model and tested the model for the Contiguous United States (CONUS) region. The authors compared the one-way and two-way coupling schemes and found that the two-way coupling scheme outperformes one-way coupling scheme. In general, the idea (developing this model – coupling scheme) is relevant for the hydrological and crop modelling community, however, the results and description of the model need to be improved.**

**Main comments**

**The description of the two-way coupling scheme is not clear. A description of the study area, relevant information about the study area (especially for hydrological and crop modeling data), and model calibration/optimization technique are needed. The model results do not convince me, simulated crop yields (Figure 2) look very different with observed.**

Thank you for your comments and suggestions. In the Introduction section, we have provided relevant information about the study area. We decided not to create a separated section for the study area description because our primary focus is on developing a coupled framework that can be applied globally. To demonstrate the framework's effectiveness, we selected the CONUS (Continental United States) for testing. This region was chosen because it offers data on both yield and irrigation water withdrawals, and comprise a variety of hydroclimatic conditions allowing for a thorough evaluation of the model. However, it is important to note that our framework is designed to be flexible and applicable to different regions worldwide. Hence, the specific study area is less central to our overall objectives, as our primary goal is to develop a globally applicable model.

For hydrological modeling, we used data from the PCR-GLOBWB 2 model, which has been extensively tested and validated at both local to global scales. On the crop modeling side, we used the WOFOST model, which has been similarly tested, with parameters fine-tuned across various regions and climatic conditions. For model calibration, we utilized the parameters provided by the respective models (PCR-GLOBWB 2 and WOFOST), as these parameters have been fine-tuned and extensively tested across multiple scales, from local to global.

Regarding Figure 2, the differences between simulated and observed yields can be attributed to the fact that the observed yields show a clear trend across the three crops both in irrigated and rainfed systems. However, we don't see such a trend present in the simulated yields. As described in the text (lines 593-598 in the manuscript with track changes), the temporal evolution in observed yields is primarily attributed to

technological advancements, which includes improved agricultural practices and introduction of enhanced crop varieties over the study period.

This explains the differences observed over time. To ensure a meaningful and fair comparison, we have selected a period where yield trends appear more stable and better aligned with the simulated yields. For the selected periods, we think that the results are convincing, and, except for rainfed, Soybean, they are certainly up to par with the results from other crop growth modelling studies at continental scales.

Additionally, to provide more clarity on the coupling scheme, we have added the schematic figures (now shown as Figure 3 in the revised manuscript, lines 426-432) that offer a detailed visualization of one-way and two-way coupling approaches. We have also enhanced the description of coupling schemes in the methodology section. Furthermore, we have addressed reviewer comments in detail and improved the text in the manuscript accordingly.

**Detail comments:**

1) **Figure 2 could be improved for understanding the figure technically and the text L226-281 regrading the coupling schemes. I would be interested in seeing the figure showing the conceptual model PCR-GLOBWB 2 and WOFOST with their components, water fluxes, and exchange variables/fluxes between these two models. This would be very helpful for understanding the coupling schemes as well as for understanding the text.**

Figure 1 (which I believe was mistakenly referred to as Figure 2) has been improved and now shows schematics of the model structure of PCR-GLOBWB 2 and WOFOST (referred to as Figure 1, lines 194-202 in the manuscript with track changes). Additionally, we have added Figure 3 (lines 426-432 in the manuscript with track changes), that illustrates the schematic view of coupling approaches along with the variables exchange. The coupling schemes of one-way and two-way text (L226-281 in the original manuscript) are provided with the details to better understand the coupling (now referred to as lines 433-500 in the manuscript with track changes).

2) **L186: Where did the model get water for irrigation (from groundwater, river, or reservoir)?**

Depending on availability, PCR-GLOBWB 2 sources water from surface water (rivers and reservoirs), groundwater (both renewable and non-renewable) and desalinated water. This is updated in the revised text in the PCR-GLOBWB 2 model description (lines 260-263 in the manuscript with track changes)

3) **L251: "...of the meteorological variables and..." change to "...of the meteorological variables at the current time step and..."?**

Line 251 in the original manuscript has been revised and is now updated as line 467 in the manuscript with track changes.

**4) L250-253: As I understand, this step is to calculate potential evapotranspiration (ETp) not actual evapotranspiration (ETa). So, where does actual Eta (L257) come from?**

WOFOST calculates both potential evapotranspiration and actual evapotranspiration. At the start of the day, PCR-GLOBWB 2 passes the previous day's soil moisture to the WOFOST, assuming no root development has occurred overnight. WOFOST then computes the potential evapotranspiration based on the meteorological variables at the current time step and the pertinent vegetation states from the previous time step (leaf area index (LAI), rooting depth, and crop height). It also calculates the actual bare soil evaporation, actual transpiration (actual evapotranspiration), potential evaporation and open water evaporation; The revised text can be found in lines 464-470 in the manuscript with track changes

**5) L267-268: "…is aggregated to the average value…" is it the summation or the average value? I think the we should pass the summation of soil moisture from the two layers of the PCR-GLOBWB 2 to the WOFOST model to ensure the amount of water is the same because the soil moisture is not reliable if the soil layer of two models have different depth and porosity.**

Apologies for the confusion. This has been clarified in the revised text lines 485-487 in the manuscript with track changes.

**6) L269:270: "WOFOST computes the actual transpiration…". Was actual transpiration already calculated (including in the actual EVAPOtranspiration term in L257).**

At L257 (original manuscript), at the start of the day, WOFOST calculates the actual transpiration (fluxes) using the previous day's soil moisture from PCR-GLOBWB 2. Then, WOFOST passes the calculated fluxes to PCR-GLOBWB 2. The imposed fluxes in PCR-GLOBWB 2 are used to update the soil moisture content, which is then sent back to WOFOST at the end of the day to calculate new fluxes for the next day.

The L269-270 describes the computation of actual transpiration (fluxes) in WOFOST for the next day.

This is clarified in the updated text lines 488-491 in the manuscript with track changes.

**7) It is not clear to me which technique the authors used for model calibration/parameter optimization**

We did not use any specific technique for the calibration or parameter optimization of the models in this study. Both PCR-GLOBWB 2 and WOFOST models have been extensively validated and tested across a wide range of scales, from local to global.

PCR-GLOBWB 2 is in principle not calibrated or tuned, although we have made subjective choices regarding the parameterization. But overall, the model is not calibrated. While,

WOFOST has been finely tuned to account for diverse climate and soil conditions, particularly for commonly studied crops such as maize, soybean, and wheat, thereby, reducing the need for further recalibration. This pre-tuning ensures that simulations reliably capture the growth and yield responses of these crops under varying environmental conditions. The updated text on the fine-tuning of crop variables and their documentation can be found in the WOFOST description section, lines 290-294 in the manuscript with track changes.

8) **Please show a figure and describe the study area somewhere before the Results section**

In the introduction section, we have provided relevant information about the study area. We decided not to create a separate section for the study area description because our primary focus is on developing a coupled framework that can be applied globally. To demonstrate the framework's effectiveness, we selected the CONUS (Continental United States) for testing. This region was chosen because it offers data on both yield and irrigation water withdrawals, as well as a variety of hydroclimatic conditions present, allowing for a thorough evaluation of the model. However, it is important to note that our framework is designed to be flexible and applicable to different regions worldwide. The specific study area is less central to our overall objectives, as our primary goal is to develop a globally applicable model.

9) **L376:378: "our coupled PCR-GLOBWB 2 – WOFOST model framework simulated yields do not capture such trends, as the modelling approach intentionally omitted to incorporate trends in technology and management practices": Please explain why? If the**

This intentional omission was to focus on the intrinsic biophysical processes and climatic conditions affecting crop yields, providing a baseline understanding unaffected by external advancements. The revised text can be found in lines 598-600 in the manuscript with track changes.

10) **L373-375: "This temporal evolution is primarily attributed to technological advancements, encompassing improved agricultural practices and the introduction of enhanced crop varieties over the study period" I did not see any trend from 1979-2007 (soybean and wheat yield in rainfed crops – Figure 2) but the model still cannot has a good match?**

The trends in reported yields differ significantly across all crops and between irrigated and rainfed systems. For maize, both irrigated and rainfed yields show an increasing trend, particularly post-2000, which is not reflected in the simulated yields. Soybean yields exhibit a gradual upward trend in irrigated systems, while rainfed soybean yields show little to no discernible trend until 2007, followed by a slight increase. Wheat yields, both irrigated and rainfed, demonstrate fluctuations with a slight upward trend towards the end of the period. These discrepancies can be attributed to various factors, including technological advancements, improved agricultural practices, and the introduction of enhanced crop varieties, which were not incorporated into the modelling approach. To

ensure a consistent and meaningful analysis, we selected the years 2006-2019 for further analysis (spatial analysis (Figure 5 in the manuscript with track changes) and evaluation metrics (Table. 1)). This period was selected because reported yields during these years appear more stable and are better aligned with the simulated yields, allowing for a fair evaluation of the model's accuracy and reliability. For the selected periods, we think that the results are convincing, and, except for rainfed, Soybean, they are certainly up to par with the results from other crop growth modelling studies at continental scales. The text is revised and updated in the result section, lines 601-613 in the manuscript with track changes.

---

## Author Response (AR2)

**Response to reviewer comments**

Dear Editor,

Thank you very much for handling our manuscript. We sincerely thank the second anonymous referees for the time, valuable comments, and suggestions, which have helped us to improve our manuscript. We have carefully revised the manuscript to include the inputs from the referee and are attaching the updated version. Please find below a point-by-point response to the reviewer comments. We hope that these revisions make our work acceptable for publication in HESS.

**Response to Reviewer comments**

We thank the Referee for the thorough review, which has helped to improve our paper.

**The authors have made commendable efforts to address the concerns raised during the first review. The expanded Introduction and Methodology sections, revised figures, and clarified details regarding the coupling framework reflect an attempt to improve the manuscript.**

**However, significant concerns remain regarding the interpretation of results. Below, I outline specific issues that need further attention:**

Thank you for your comments regarding the interpretation of the results. We have carefully re-examined our results and identified a few aspects that required attention.

First, we noticed that the WOFOST standalone model was not harmonized with the soil properties when compared to the coupled models. Previously, the standalone model used spatially constant soil properties, which may have affected comparability. Second, we observed that WOFOST does not account for the residual moisture content, whereas PCR-GLOBWB 2 does. To ensure consistency, we have adjusted the models accordingly and rerun the simulations with harmonized soil properties across all models. Additionally, we have updated the methodology section to reflect these harmonized efforts (lines 521-526 in the manuscript with track changes).

Furthermore, we have rewritten our rationale and hypothesis in the Introduction section (please refer to lines 96-103; lines 132-146 in the manuscript with track

changes). We have also updated the results and the corresponding text accordingly. Please refer to the manuscript with track changes for details.

1. **Lines 591–605: The authors claim that two-way coupling improves hydrological simulations for rainfed crops by incorporating soil moisture dynamics and detailed processes. However, the normalized RMSE for rainfed maize in the stand-alone model (0.22) is notably smaller than in the two-way coupled model (0.50) (Table 1). This discrepancy contradicts the claim of improved performance and raises questions about the validity of the conclusions. The authors need to explicitly discuss whether this result is due to calibration issues, model assumptions, or inherent limitations of the coupling approach, particularly if the goal is to emphasize scientific contributions.**

Thank you for this observation. The model performance metrics table (Table 1) was previously filled with an erroneous value for the two-way coupling, resulting in identical values being reported for both the one-way and two-way models. We have updated the validation section and corrected the values in Table 1 based on the new model runs with harmonized soil properties (see updated section 3.2 in the manuscript with track changes).

2. **Lines 722–733: The added text claims that two-way coupling captures crop stress feedback mechanisms that are missing in one-way coupling, explaining regional yield differences. However, these claims lack supporting evidence, such as observational validation or references to previous studies. To address this, the authors should provide supporting references or additional quantitative analyses, and clearly distinguish between conclusions based on results and those based on assumptions or hypotheses. Furthermore, the differences in feedback mechanisms between the coupled and stand-alone models are not adequately explained and require clarification.**

Thanks for the comment. We have added a new section to the supplementary material, referenced in the main text, to further illustrate the feedback mechanism between one-way and two-way coupling. This section highlights how soil moisture dynamics respond to crop growth and capture crop water stress. Please refer to lines 803-807 in the manuscript with track changes. Additionally, we have included a detailed explanation of the differences in feedback mechanisms between the coupled model and the standalone model. Please see

lines 634-689 in the manuscript with track changes and lines 58-346 in the supplementary information with the updated version.

3. **Lines 699–709: The claim that the stand-alone model overpredicts yields under rainfed conditions is not supported by direct observational comparisons, and is therefore speculative. Moreover, this conclusion seems inconsistent with the time-series data in Figure 4.**

We have revised the text and referred to the figures where necessary. Please see the updated text in lines 790-801 in the manuscript with track changes.

4. **While Figures 6–9 have been revised, the interpretation of results, such as "notably" (e.g., lines 701, 710), is not convincingly supported. If differences are quantitatively significant, they should be more clearly highlighted, e.g., by bar plots.**

Thanks for the comment. We have added Supplementary Figures S13 and S14. (see Supplementary Information V) to supplementary document showing the relative difference in 1979-2019 mean and coefficient of variation between two-way coupling and stand-alone runs for rainfed maize, soybean, and wheat crops. We refer to these newly developed figures in line 796 in the manuscript with track changes.

We have referred to the figures where necessary to support the statement (line 783; line 803 in the manuscript with track changes)